# LABEL-CONSISTENT CLUSTERING FOR EVOLVING DATA

## ABSTRACT

Data analysis often involves an iterative process, where solutions must be continuously refined in response to new data. Typically, as new data becomes available, an existing solution must be updated to incorporate the latest information. In addition to seeking a high-quality solution for the task at hand, it is also crucial to ensure consistency by minimizing drastic changes from previous solutions. Applying this approach across many iterations, ensures that the solution evolves gradually and smoothly.

In this paper, we study the above problem in the context of clustering, specifically focusing on the $k$-center problem. More precisely, we study the following problem: Given a set of points $X$, parameters $k$ and $b$, and a prior clustering solution $\mathcal{H}$ for $X$, our goal is to compute a new solution $\mathcal{C}$ for $X$, consisting of $k$ centers, which minimizes the clustering cost while introducing at most $b$ changes from $\mathcal{H}$. We refer to this problem as *label-consistent $k$-center*, and we propose two constant-factor approximation algorithms for it. We complement our theoretical findings with an experimental evaluation demonstrating the effectiveness of our methods on real-world datasets.

## 1 INTRODUCTION

Data clustering is a fundamental problem in data analysis with numerous applications across various domains (Gan et al., 2020). Traditionally, clustering is studied in a one-shot setting, where the goal is to find an optimal solution for a given dataset without incorporating prior information. However, in many applications, clustering is often an iterative process. Instead of starting from scratch, a clustering solution from a previous computation may be available. The objective then becomes to *refine* that solution in order to be *consistent* with newly-introduced or modified data. In such a scenario, we are interested in finding a high-quality solution, which is also *close enough* to the previously-available solution, so as to preserve stability and continuity.

As an example, consider clustering a stream of evolving data, such as news articles. To effectively monitor major news stories, we want to update our clustering daily as new articles arrive. However, rather than forming entirely new clusters each day, we seek to maintain continuity with the previous day's clustering. This approach is crucial because major news stories often unfold over multiple days. By ensuring consistency in clustering, we can track evolving narratives more effectively, preserving the coherence of long-running stories.

In this paper, we study the problem of refining an existing (but possibly obsolete) clustering in the presence of new data, while respecting a consistency requirement between the refined solution and the existing one.

More specifically, we introduce a novel problem, which we call *label-consistent $k$-clustering*, defined as follows. Given a set of points $X$, parameters $k$ and $b$, and a prior clustering solution $\mathcal{H}$ for $X$, our goal is to compute a new solution $\mathcal{C}$ for $X$, consisting of $k$ centers and minimizing the clustering cost while introducing at most $b$ changes to $\mathcal{H}$. The type of changes that we account for is *labeling changes*, namely, data points that are re-assigned to a different cluster; see Definition 3 for a formal definition. We focus on consistency in the context of the classic $k$-center problem Gonzalez (1985), to define *label-consistent $k$-center*.

We refer to the prior solution $\mathcal{H}$ as *historical clustering*. Our formulation is agnostic to how the historical clustering $\mathcal{H}$ has been computed, we only assume that the data points in $X$ can be assigned to centers in $\mathcal{H}$. The problem we study is motivated by scenarios where the historical clustering is *fixed and immutable*, that is, it has already been presented to the user (e.g., a set of clients assigned to services, which has already been published) and cannot be retracted. In such a case, even if the historical clustering is imperfect, in order to avoid disruptions we want to require that the new clustering stays consistent to the historical clustering.

In the case of clustering evolving data, our framework can be instantiated as follows. At time $t$ a clustering solution $\mathcal{C}_t$ is computed recursively on data $X_t$, using solution $\mathcal{C}_{t-1}$ as historical clustering, where at time $0$ any standard clustering method can be used to compute $\mathcal{C}_0$. At time $t$, we want to refine our clustering solution for data $X_t$, which may have additions and/or deletions with respect to $X_{t-1}$. For clarity, first consider the case when $X_t$ contains only new points, i.e., no deletions. In that case, we extend the solution $\mathcal{C}_{t-1}$ by assigning the newly added points to their closest centers. Now if $X_t$ deletes some points from $X_{t-1}$, we can simply remove them from the cluster assignment in solution $\mathcal{C}_{t-1}$, provided they were not centers in $\mathcal{C}_{t-1}$. Otherwise, for each deleted point $p$ that was a center in $\mathcal{C}_{t-1}$, we select another point from $\mathcal{C}_{t-1}$ (that is still present in $X_t$) as a new cluster center. This ensures that all remaining points in the cluster centered at $p$ in $\mathcal{C}_{t-1}$ are still present together in a cluster with a new center.[1]

We can now apply our *label-consistent $k$-clustering* problem and seek a new solution $\mathcal{C}_t$ that minimizes the clustering cost while relabeling at most $b$ data points in $X_t$ with respect to their current assignment to centers in $\mathcal{H}$. This captures the essence of $\mathcal{H}$, while allowing us to derive insights from the new data.

Prior work has considered similar frameworks for ensuring stability between clustering solutions, including a different formulation of consistent $k$-clustering (Lattanzi and Vassilvitskii, 2017), notions of evolutionary clustering (Chakrabarti et al., 2006), and a recent work on resilient $k$-clustering (Ahmadian et al., 2024). More discussion on the differences of those approaches from our formulation is provided in the related-work section, while an empirical evaluation with the resilient $k$-clustering approach is presented in our experiments.

Given that the problem we study is **NP**-hard, we present two constant-factor approximation algorithms for *label-consistent $k$-center*. The first algorithm, named OVERCOVER, is a tight 2-approximation algorithm, albeit running in $2^k\, poly(n)$. Our second algorithm, named GREEDYANDPROJECT, is a 3-approximation algorithm that runs in polynomial time and serves as the main result of the paper. In summary, in this paper we make the following contributions.

- We introduce the problem of *label-consistent $k$-clustering*, a novel clustering formulation that aims to optimize clustering cost while ensuring a consistency constraint, in the form of maximum number of data point re-labelings, from a historical clustering.
- We present two constant-factor approximation algorithms for the $k$-center variant of the proposed problem (Theorem 2 and Theorem 3).
- We present a thorough experimental evaluation on real-world datasets and on different settings, comparing our algorithms with standard $k$-center algorithms and a state-of-the-art baseline.

## 2   RELATED WORK

Data clustering is a widely-studied topic (Gan et al., 2020). Most related to our paper are methods with provable approximation guarantees. In this area, research has focused on problems, such as $k$-means (Ahmadian et al., 2019), $k$-median (Charikar et al., 1999), and $k$-center (Gonzalez, 1985). For the $k$-center problem, which is the focus of this paper, the classic CARVE algorithm (Hochbaum and Shmoys, 1985) provides a 2-approximation. This algorithm is a recurrent subroutine for the methods we develop in this paper.

---

[1]Note that if all the points of the cluster centered at $p$ in $\mathcal{C}_{t-1}$ are deleted in $X_t$, then we can simply remove the entire cluster from $\mathcal{C}_{t-1}$.

Traditional methods do not provide consistency requirements, and thus, a small change in the data might result in a large discrepancy on the clustering result. Since consistency is a desirable property in many applications, researchers have also considered formulations to account for this requirement.

**Evolutionary clustering.** Different definitions have been proposed to measure temporal smoothness for clustering evolving data. Chakrabarti et al. (2006) introduce a general framework where they impose constraints on the changes of clustering solutions between consecutive snapshots. Chi et al. (2009) present a spectral approach, where the temporal-smoothness component of their approach accounts for cluster membership. These approaches, as well as followup research (Folino and Pizzuti, 2013; Xu et al., 2014), present mainly heuristic methods that do not offer quality guarantees.

**Dynamic clustering.** There is extensive work in the clustering literature proposing different formulations and methods to minimize the number of updates (recourse) for evolving data (Chan et al., 2018; Pellizzoni et al., 2023; Bateni et al., 2023). However, existing approaches differ fundamentally from ours. These dynamic algorithms treat label-consistency as a soft objective to be minimized, either per update or across updates for amortized cost, rather than as a hard constraint enforced in the problem definition. Furthermore, our problem is motivated by a regime where the number of points is extremely large compared to the number of time steps in which the input changes. This stands in contrast to typical dynamic settings, which often assume many updates and therefore require sublinear or near-sublinear running times.

**Low-recourse algorithms.** More generally, low-recourse algorithms have been proposed for problems other than clustering, and share similar goals with our approach. For instance,Bhattacharya et al. (2023) consider online bipartite matching with constraints on the amortized number of replacements, achieving logarithmic recourse bounds. Similarly, Bernstein et al. (2019) investigate the problem of maintaining solutions to time-varying optimization problems while minimizing the cost of transitions between solutions. Similarly to our difference with dynamic clustering, this line of work typically treats recourse as a soft objective to be minimized or bounds it in an amortized sense, whereas recourse is a hard constraint in our problem definition.

**Consistent clustering (a different definition).** Lattanzi and Vassilvitskii (2017) introduce the idea of *consistent clustering*, but their notion of consistency is significantly different than ours. In particular, they measure consistency in terms of the symmetric differences between the sets of cluster centers, while we measure changes in the assignment of data points to centers. We elaborate on the difference with our definition in Section 3. Furthermore, Lattanzi and Vassilvitskii (2017) and followup research (Fichtenberger et al., 2021; Łącki et al., 2024; Forster and Skarlatos, 2025) consider an online setting and provide competitive-ratio analysis.

**Resilient clustering.** More recently, Ahmadian et al. (2024) introduced the notion of *resilient $k$-clustering*. Their goal is to find a clustering that is resilient to perturbations in the input data. Their resiliency definition asks to preserve cluster membership of the data points, similarly to our consistency definition. The main difference is that they aim to achieve resiliency *without any prior knowledge to historical clustering*; this makes their setting more strict, and as we will see in our experimental evaluation, their algorithm often gives poor results in terms of resiliency/consistency measure. This line of research falls in the topic of *perturbation resilience* (Balcan and Liang, 2016; Chekuri and Gupta, 2018; Bandyapadhyay, 2022), where the goal is to find an optimal clustering that does not change when the data are perturbed by a small amount. Instead, we do not make any assumption about the input data.

## 3 LABEL-CONSISTENT $k$-CLUSTERING

In this section, we introduce our notation and present the formal definition of the clustering problem we study.

**Clustering.** Let $(X, d)$ be a metric space on $n$ points. For any $Y \subseteq X$ and $x \in X$, we use $d(x, Y)$ to denote the minimum distance from $x$ to any point in $Y$, that is, $d(x, Y) = \min_{y \in Y} d(x, y)$. A $k$-clustering of a set

of points $X$ is a pair $\mathcal{C} = (C, \ell_C)$, where $C \subseteq X$ is a set of $k$ *cluster centers*, and $\ell_C : X \mapsto C$ is *labeling function*, which assigns each point $x \in X$ to a center $\ell_C(x) \in C$.

**Definition 1** (The $k$-clustering problem). *Fix some positive real $p \in \mathbb{R}_{\geq 1}$. Given a metric space $(X, d)$ and a positive integer $k \in \mathbb{Z}_+$, the $k$-Clustering problem asks to find a $k$-clustering $\mathcal{C} = (C, \ell_C)$ of $X$ that minimizes the cost function $\mathrm{cost}_p(\mathcal{C}) = \left( \sum_{x \in X} d(x, \ell_C(x))^p \right)^{1/p}$.*

When $p = 1$, the objective yields the *$k$-median* problem, and when $p = 2$, it captures the *$k$-means* objective. When $p \to \infty$, it recovers the *$k$-CENTER* problem, where $\mathrm{cost}_\infty(\mathcal{C}) = \max_{x \in X} d(x, \ell_C(x))$.

**Label-consistent clustering.** To present the proposed notion of label-consistent clustering, we start by defining the distance between two clusterings.

**Definition 2** (Distance between two clusterings). *Let $\mathcal{C}_1 = (C_1, \ell_1)$ and $\mathcal{C}_2 = (C_2, \ell_2)$ be two $k$-clusterings over the same set of points $X$. We define the distance between $\mathcal{C}_1$ and $\mathcal{C}_2$ as $\Delta(\mathcal{C}_1, \mathcal{C}_2) = |\{x \in X, \ell_1(x) \neq \ell_2(x)\}|$, namely, the number of points in $X$ that are assigned to different cluster centers in $\mathcal{C}_1$ and $\mathcal{C}_2$.*

In this work, we measure the distance between two clusterings $\mathcal{C}_1$ and $\mathcal{C}_2$ using the distance function $\Delta(\mathcal{C}_1, \mathcal{C}_2)$ of Definition 2. An alternative definition is the size of the symmetric difference $|C_1 \triangle C_2|$, which was used to define consistent clustering in online settings (Lattanzi and Vassilvitskii, 2017). One can observe that the two clustering distance functions $\Delta(\mathcal{C}_1, \mathcal{C}_2)$ and $|C_1 \triangle C_2|$ can take very different values.

To ensure consistency with an available historical clustering $\mathcal{H}$, we use a parameter $b$ to control the degree of label consistency in the resulting clustering, and we require to find a $k$-clustering $\mathcal{C}$ that has minimal clustering cost while satisfying the consistency constraint $\Delta(\mathcal{H}, \mathcal{C}) \leq b$. More formally, we define the following problem.

**Definition 3** (LABEL-CONSISTENT $k$-CLUSTERING). *Fix some $p \in \mathbb{R}_{\geq 1}$. Given a metric space $(X, d)$, two positive integers $k, b \in \mathbb{Z}_+$, and a historical clustering $\mathcal{H}$ of $X$ with $\bar{k}$ centers, the LABEL-CONSISTENT $k$-CLUSTERING problem seeks to find a $k$-clustering $\mathcal{C} = (C, \ell_C)$ of $X$ that minimizes the cost function $\mathrm{cost}_p(\mathcal{C}) = \left( \sum_{x \in X} d(x, \ell_C(x))^p \right)^{1/p}$, while ensuring $\Delta(\mathcal{H}, \mathcal{C}) \leq b$.*

In this paper we focus on the $\mathrm{cost}_\infty(\cdot)$ objective, namely the *$k$-center* objective.

**Definition 4** (LABEL-CONSISTENT $k$-CENTER). *The LABEL-CONSISTENT $k$-CENTER problem is the instantiation of the LABEL-CONSISTENT $k$-CLUSTERING problem with $p = \infty$, and thus, $\mathrm{cost}_\infty(\mathcal{C}) = \max_{x \in X} d(x, \ell_C(x))$.*

We denote by $\mathcal{I} = ((X, d), k, \mathcal{H} = (H, \ell_H), b)$ an instance of LABEL-CONSISTENT $k$-CENTER. We say a (clustering) solution $\mathcal{C} = (C, \ell_C)$ to $\mathcal{I}$ is *feasible* if $|C| = k$ and $\Delta(\mathcal{H}, \mathcal{C}) \leq b$, i.e., $\mathcal{C}$ opens $k$ centers and reassigns at most $b$ points from the historical clustering $\mathcal{H}$. We denote by $\mathcal{C}^* = (C^*, \ell_{\mathcal{C}^*})$ as an optimal solution to $\mathcal{I}$, and denote by $r^* := \mathrm{cost}_\infty(\mathcal{C}^*)$, the optimal *radius* of $\mathcal{C}^*$.

An interesting observation is that an optimal solution $\mathcal{C}^*$ for LABEL-CONSISTENT $k$-CENTER may not necessarily assign each point in $X$ to its closest center in $\mathcal{C}^*$, as the consistency constraint may force a data point to be assigned to its historical cluster center although it is not the closest center.

It is easy to see (by setting $b = n$) that, for specific values of $p$, LABEL-CONSISTENT $k$-CLUSTERING inherits the hardness of the corresponding $k$-clustering problem. In fact, for LABEL-CONSISTENT $k$-CENTER, we obtain the following stronger hardness results,[2] whose proof is deferred to Appendix B.1.

**Theorem 1.** *For every $\epsilon > 0$ and every computable function $f$, the following hold:*

    *(i) It is **NP**-hard to approximate LABEL-CONSISTENT $k$-CENTER within a factor of $(2 - \epsilon)$.*

---

[2] Similar hardness of approximation of results follow for LABEL-CONSISTENT $k$-median and $k$-means.

*(ii)* *It is* **W[2]***-hard to approximate* LABEL-CONSISTENT $k$-CENTER *within a factor of* $(2 - \epsilon)$ *when parameterized by $k$ (Downey and Fellows, 2012). Thus, there is no $(2 - \epsilon)$-approximation algorithm running in time $f(k)\, n^{O(1)}$.*

*(iii)* *The above hardness results remain valid even if one is allowed to reassign $g(b) \geq b$ points, for any function $g$.*

## 4 ALGORITHMS

In this section, we present two constant-factor approximation algorithms for the LABEL-CONSISTENT $k$-CENTER problem. In Section 4.1, we present OVERCOVER, a tight 2-approximation algorithm that runs in FPT time in parameter $k$ (Downey and Fellows, 2012). GREEDYANDPROJECT, the main contribution of this paper, is presented in Section 4.2, and is a 3-approximation algorithm that runs in polynomial time. We remark that both of our algorithms work under the assumption that the optimal radius $r^*$ is known. In Section 4.3 we discuss how this assumption can be removed and its relation to the budget parameter $b$.

**Notation.** Let $(X, d)$ be a metric space. For a point $x \in X$ and $r \in \mathbb{R}_{\geq 0}$, we denote by $\text{Ball}(x, r)$ as the set of points that are at a distance at most $r$ from $x$. $\text{poly}(n)$ denotes a fixed function that is polynomial in $n$.

### 4.1 A TIGHT 2-APPROXIMATION IN FPT TIME

In this section, we design an FPT 2-approximation algorithm for LABEL-CONSISTENT $k$-CENTER.

**Theorem 2.** *There is a $2$-approximation algorithm for the* LABEL-CONSISTENT $k$-CENTER *problem running in time $2^k \, \text{poly}(n)$.*

**Overview of the OVERCOVER algorithm.** The high level idea of our algorithm, is simple and clean: first, we assume that we know the optimal radius $r^*$ and the historical centers $H^*$ that are present in an optimal solution; next, we identify the points that are far from $H^*$ using $r^*$ and cluster them using any classical 2-approximation $k$-CENTER algorithm; finally, we reassign points to minimize the number of reassignments. A crucial part of the analysis is to show that OVERCOVER opens at most $k$ centers.

In more detail, let $\mathcal{C}^* = (C^*, \ell_{\mathcal{C}^*})$ be an optimal solution to $\mathcal{I}$ with cost $r^*$, and let $H^* = C^* \cap H$, be the set of historical centers present in $C^*$. First, the algorithm guesses $H^*$. Next, it considers all the points in $X_B := X \setminus \cup_{h \in H^*} \text{Ball}(h, r^*)$, and clusters them using CARVE algorithm of Hochbaum and Shmoys (1985). The CARVE algorithm, when given a dataset $X'$ and radius $r$, repeatedly picks a center $s' \in X'$ and deletes all the points within distance $2r$ from $s'$, until all the points are deleted. If $S'$ is the set of picked centers by CARVE, then it is easy to see that $d(x', S') \leq 2r$ for every $x' \in X'$. Finally, the sets $S'$ and $H^*$ are merged to obtain a center set $C$, and points are assigned to $C$ minimizing the number of reassignments. Pseudocode for OVERCOVER and CARVE are available in Appendix A, and the analysis is deferred to Appendix B.2.

### 4.2 A 3-APPROXIMATION ALGORITHM

In this section, we present a polynomial time 3-approximation algorithm for LABEL-CONSISTENT $k$-CENTER.

**Theorem 3.** *There is a $3$-approximation algorithm for the* LABEL-CONSISTENT $k$-CENTER *problem running in time $O(n^2 \log n + nk \log n)$.*

The presentation proceeds as follows: we outline our algorithm GREEDYANDPROJECT in Section 4.2.1 and, prove its correctness in Section 4.2.2. The proof of Theorem 3 is present in Appendix B.5.

**Algorithm 1** GREEDYANDPROJECT

---

**Input:** an instance $\mathcal{I} = ((X, d), k, \mathcal{H} = (H, \ell_H), b)$ of LABEL-CONSISTENT $k$-CENTER, optimal radius $r^*$

**Output:** a 3-approximate solution $\mathcal{C} = (C, \ell_C)$ to $\mathcal{I}$
1: $C_0, C_1, C \leftarrow \emptyset$
2: $S \leftarrow$ CARVE$(X, 2r^*)$        % Algorithm 4
3: **for** $s \in S$ **do**
4:      **if** $N_H(s) \neq \emptyset$ **then**
5:         add the maximum weight center, $\tilde{s}$, from $N_H(s)$ to $C_0$ by breaking ties arbitrarily
6:      **else** add $s$ to $C_0$
7: Let $C_1$ be the $(k - |C_0|)$ maximum weight historical centers from $H \setminus C_0$
8: Let $C \leftarrow C_0 \cup C_1$
9: **for** $x \in X$ **do**        % assign points to $C$
10:      **if** $\ell_H(x) \in C$ and $d(x, \ell_H(x)) \leq r^*$ **then** $\ell_C(x) = \ell_H(x)$
11:      **else** $\ell_C(x) \leftarrow \arg\min_{c \in C} d(x, c)$
12: **return** $\mathcal{C} = (C, \ell_C)$

---

**Algorithm 2** Algorithm to analyze Algorithm 1

---

**Input:** $\mathcal{I} = ((X, d), k, \mathcal{H} = (H, \ell_H), b)$, optimal solution $\mathcal{C}^* = (C^*, \ell_{\mathcal{C}^*})$, set $S$ obtained from Line 2 of Algorithm 1

**Output:** a feasible solution $\mathcal{C}' = (C', \ell_{C'})$ to $\mathcal{I}$
1: $C_0', C_1', C_2', C' \leftarrow \emptyset$
2: **for** $s \in S$ **do**
3:      **if** $N_H(s) \neq \emptyset$ **then**
4:         add the maximum weight center, $\hat{s}$, from $N_H(s)$ to $C_0'$ by breaking ties arbitrarily
5:      **else** add $s$ to $C_0'$
6: **for** $s \in S_\gamma$ **do**     % $S_\gamma = \{s \in S | \Gamma_s^* \neq \emptyset\}$
7:      pick $|\Gamma_s^*| - 1$ maximum-weight historical centers, $\hat{\Gamma}_s$, from $\Gamma_s^* \setminus \{\hat{s}\}$
8:      $C_1' \leftarrow C_1' \cup \hat{\Gamma}_s$
9: Let $C_2'$ be the $|H_f^*|$ maximum weight historical centers from $H \setminus (C_0' \cup C_1')$
10: Let $C' \leftarrow C_0' \cup C_1' \cup C_2'$
11: **for** $x \in X$ **do**
12:      **if** $\ell_H(x) \in C$ and $d(x, \ell_H(x)) \leq r^*$ **then** $\ell_C(x) = \ell_H(x)$
13:      **else** $\ell_C(x) \leftarrow \arg\min_{c \in C} d(x, c)$
14: **return** $\mathcal{C}' = (C', \ell_{C'})$

---

### 4.2.1 OVERVIEW OF THE GREEDYANDPROJECT ALGORITHM

The pseudo-code of our GREEDYANDPROJECT algorithm is described in Algorithm 1. We assume that the algorithm has access to the optimal radius $r^*$ of the input instance $\mathcal{I} = ((X, d), k, \mathcal{H} = (H, \ell_H), b)$. Let the historical clusters be denoted as $\hat{\Pi} = \{\hat{\pi}_h\}_{h \in H}$, where $\hat{\pi}_h = \{x \in X, \ell_H(x) = h\}$. For a historical center $h \in H$, the *weight* of $h$, denoted by $w(h)$, is the total number of points in Ball$(h, r^*) \cap \hat{\pi}_h$. For $x \in X$, let $N_H(x)$ denote the set of historical centers within distance $r^*$ from $x$, i.e., $N_H(x) = \{h \in H | d(x, h) \leq r^*\}$. For a subset $T \subseteq H$ of historical centers, let $w(T) = \sum_{h \in T} w(h)$. Let $\mathcal{C}^* = (C^*, \ell_{\mathcal{C}^*})$ be a fixed (but unknown) optimal solution to $\mathcal{I}$. The algorithm works in three phases:

($i$) **Greedy phase:** In this phase (Line 2), the algorithm computes a set $S$ of centers such that, for every point $x \in X$ it is $d(x, S) \leq 2r^*$ (see CARVE Algorithm 4), $|S| \leq k$, and $d(s, s') > 2r^*$, for $s \neq s' \in S$.

($ii$) **Project phase:** In this phase (**for**-loop in Line 3), for every $s \in S$, the algorithm swaps $s$ with a maximum weight historical center in $N_H(s)$, if it is non-empty. Note that, if $s$ is served by a historical center in $\mathcal{C}^*$, then there exists a historical center in $N_H(s)$.

($iii$) **End phase:** In this phase (Lines 7-12), the algorithm first adds $(k - |S|)$ historical centers of maximum weight that have not yet been picked to its center set $C$ so that $|C| = k$. Then, it assigns every point $x \in X$ to $C$ minimizing the number of reassignments.

It is easy to see that, for every $x \in X$, it holds that $d(x, C) \leq 3r^*$. However, the crucial part is to show that the number of reassignments by the algorithm is at most $b$, i.e., $\Delta(\mathcal{H}, \mathcal{C}) \leq b$. The key observation for proving this claim is that in the Project phase, if $s \in S$ is served by a historical center $h \in C^*$, then $h$ is a

candidate for swapping $s$. Furthermore, since the algorithm picks the maximum-weight historical center $\tilde{s}$, it holds that $w(\tilde{s}) \geq w(h)$. This inequality holds for every $s \in S$ as the set of centers that are within distance $r^*$ from $s \neq s' \in S$ are disjoint since $d(s, s') > 2r^*$. Finally, the algorithm adds $(k - |S|)$ historical centers of maximum weight from the remaining historical centers. Therefore, at every step, with respect to the number of reassignments, the algorithm does at least as good as the optimal solution.

### 4.2.2 ANALYSIS

In this section, we prove the guarantees of the GREEDYANDPROJECT algorithm (Algorithm 1).

**Theorem 4.** *Given an instance $\mathcal{I} = ((X, d), k, \mathcal{H} = (H, \ell_H), b)$ of* LABEL-CONSISTENT $k$-CENTER *with optimal cost $r^*$, Algorithm 1 returns a feasible $3$-approximate solution $\mathcal{C} = (C, \ell_C)$ to $\mathcal{I}$ in time $O(nk)$.*

*Proof.* First note that $|C| = k$, since we have that $|C_0| = |S| \leq k$. Furthermore, for every $x \in X$, we have that $d(x, S) \leq 2r^*$, and hence $d(x, C) \leq d(x, S) + r^* = 3r^*$. Note that this concludes the proof, as either $\ell_H(x) \in C$ and $d(x, \ell_H(x)) \leq r^*$, then $d(x, \ell_C(x)) = d(x, \ell_H(x)) \leq r^*$. Otherwise, $d(x, \ell_C(x)) = d(x, C) \leq 3r^*$ as desired. Therefore, we only need to show that $\Delta(\mathcal{C}, \mathcal{H}) \leq b$.

**Correctness:** We start with some basic definitions. Recall that, for a historical center $h \in H$, the weight of $h$ is $w(h) = |\text{Ball}(h, r^*) \cap \hat{\pi}_h|$. Fix an optimal solution $\mathcal{C}^* = (C^*, \ell_{\mathcal{C}^*})$ to $\mathcal{I}$. We say $H^* := C^* \cap H$, the set of historical centers present in the optimal centers $C^*$, as the *historical optimal centers*. Consider the set $S$ obtained from CARVE$(X, 2r^*)$, and $N_H(s) = H \cap \text{Ball}(s, r^*)$, for $s \in S$. Then, note that, for $s \neq s' \in S$, we have that $N_H(s) \cap N_H(s') = \emptyset$, since $d(s, s') > 2r^*$. Furthermore, we say that a historical optimal center $h \in H^*$ is *covered* by $s \in S$, if $h \in N_H(s)$. Note that a historical optimal center can be covered by at most one point in $S$, but a point in $S$ can cover multiple historical optimal centers. Hence, for $s \in S$, let $\Gamma_s^* = H^* \cap N_H(s)$ be the set of historical optimal centers covered by $s$. Next, let $H_c^* = \bigcup_{s \in S} \Gamma_s^*$ be the historical optimal centers covered by $S$, and let $H_f^* = H^* \setminus H_c^*$, which we call *far* historical optimal centers to $S$. Finally, let $S_\gamma \subseteq S$, be the set of points of $S$ that covers at least one historical optimal center, i.e. for which $\Gamma_s^* \neq \emptyset$. Then, $\sum_{s \in S} |\Gamma_s^*| = \sum_{s \in S_\gamma} |\Gamma_s^*| = |H_c^*|$.

To show that Algorithm 1 returns a feasible solution, we consider an *idealized* Algorithm 2 that is powerful and knows the optimal solution $\mathcal{C}^*$. Algorithm 2 receives as input: $(i)$ the instance $\mathcal{I}$, $(ii)$ the optimal solution $\mathcal{C}^*$, and $(iii)$ the set $S$ obtained from Line 2 of Algorithm 1. It computes three sets: $C_0'$, $C_1'$ and $C_2'$ and computes a set of centers $C = \cup_{i \in [3]} C_i'$. First, it sets $C_0' = S$, and replaces $s \in S$ with the maximum weight historical center $\hat{s} \in N_H(s)$ if $N_H(s) \neq \emptyset$ (**for**-loop in Line 2). Then, for every $s \in S_\gamma$, it adds $|\Gamma_s^*| - 1$ maximum-weight historical centers from $\Gamma_s^* \setminus \{\hat{s}\}$ to $C_1'$ (**for** loop in Line 6). Finally, in Line 9, the algorithm picks $|H_f^*|$ maximum-weight historical centers from the remaining historical centers for $C_2'$. Note that given $\mathcal{C}^*$, the algorithm can compute $H^*, H_c^*, S_\gamma$, and $\Gamma_s^*$ for $s \in S_\gamma$.

We first show that the solution of Algorithm 2 is a feasible solution to $\mathcal{I}$.

**Lemma 1.** *The solution $\mathcal{C}' = (C', \ell_{C'})$ returned by Algorithm 2 is a feasible solution to $\mathcal{I}$.*

Finally, we show that Algorithm 1 reassigns no more points than Algorithm 2, which is bounded by $b$.

**Lemma 2.** *The number of points reassigned by solution $\mathcal{C}$ of Algorithm 1 is no more than the number of points reassigned by the solution $\mathcal{C}'$ of idealized Algorithm 2.*

The proofs of Lemma 1 and Lemma 2 are deferred to Appendix B.3 and B.4, respectively.

*Running time:* The computation of set $S$ takes time $nk$, while the **for** loop takes time $k^2$. Line 7 takes $O(k \log k)$ time, and the final assignment takes $O(nk)$ times. Therefore, the resulting time of Algorithm 1 is $O(nk)$, finishing the proof of the theorem. $\qquad\square$

### 4.3 GUESSING AND VERIFYING THE OPTIMAL RADIUS

**Guessing the optimal radius $r^*$.** As discussed in the previous sections, our algorithms assume that the optimal radius $r^*$ is known and is given as input. This is a very common assumption in designing algorithms for $k$-CENTER and its variants. This assumption can be removed by iterating over all $n^2$ pairwise distances as a candidate for $r^*$, resulting in a multiplicative factor of $n^2$ in the overall running time. The running time can be sped up by sorting these distances and using binary search to find $r^*$. However, in practice, the distance aspect ratio $\Delta$, which is defined as the ratio of the maximum to the minimum distance, is often bounded polynomially in $n$. In such settings, it is standard to speed up the guessing of $r^*$ by discretizing all distances into powers of $(1 + \epsilon)$, for a small $\epsilon > 0$. This approach reduces the number of candidate radii from $n^2$ to $O(\log n/\epsilon)$, at the cost of introducing only a multiplicative $(1 + \epsilon)$ factor in the approximation.

**Verifying the guess for the optimal radius $r^*$.** Since the idea is to run our algorithms (Algorithm 1 and Algorithm 3) for every guess of $r^*$, we need to make sure that our algorithms return a *feasible* solution. Note that our algorithms return the solution corresponding to the guess of $r^*$ for which a feasible solution was found. To check the feasibility of a solution $\mathcal{C} = (C, \ell_C)$ (Line 12 of Algorithm 1 and Line 9 of Algorithm 3), the algorithm checks if $\Delta(\mathcal{C}, \mathcal{H}) \leq b$.

## 5 EXPERIMENTAL EVALUATION

We empirically evaluate our algorithms, OVERCOVER and GREEDYANDPROJECT, on real-world datasets. We use four temporal datasets: *Electric Consumption* (Hebrail and Berard, 2006), *OnlineRetail* (Chen, 2015), *Twitter* (Helwig et al., 2015), and *Uber*,[3] as well as one non-temporal dataset, *Abalone* (Nash et al., 1994). The description and preprocessing of these datasets is deferred to Appendix C.1. Below we describe the experimental setup, baselines, and practical improvements of our methods. Finally, we present our results.

### 5.1 EXPERIMENTAL SETUP

We consider four setups. In the main body of the paper, we focus on the first setup, which simulates accommodating historical clustering upon arrival of new data, and the second setup, which simulates temporal evolution of data. The third setup evaluates our algorithms on noisy data, and is presented in Appendix C.2, and the fourth setup explores scalability with respect to input parameters, as presented in Appendix C.3.

**Baselines.** We evaluate our algorithms against three baselines:

(i) CARVE algorithm Hochbaum and Shmoys (1985), described in Algorithm 4.

(ii) FFT algorithm Gonzalez (1985), which starts with an arbitrary center, and repeats the following operations $k - 1$ times: find the point furthest away from the current set of centers and add it to the set of centers.

(iii) Resilient $k$-clustering. We implemented a version of the algorithm described by Ahmadian et al. (2024). This algorithm first opens up to $\alpha k$ centers in a resilient way, and then $\beta k$ centers using FFT, opening in total up to $(\alpha + \beta)k$ centers. To ensure a fair comparison, we set $\alpha = \beta = 0.5$, to open $k$ centers. We refer to this algorithm as RESILIENT.

Next, we detail the first two setups. We denote by HIST a historical clustering algorithm and ALG an algorithm for LABEL-CONSISTENT $k$-CENTER.

**Setup 1: New data arrival.** Consider a dataset $X$. First, we apply FFT on $X$ to find $3k$ clusters $X = C_1 \cup \ldots \cup C_{3k}$. Then, we consider the dataset $X' = C_1 \cup \ldots \cup C_k \subseteq X$. We run HIST on $X'$ and take the result to be the historical clustering $\mathcal{H}$, and define $\ell_H(x)$ for every $x \in X \setminus X'$ to be the closest center from $x$

---

[3] https://www.kaggle.com/datasets/fivethirtyeight/uber-pickups-in-new-york-city

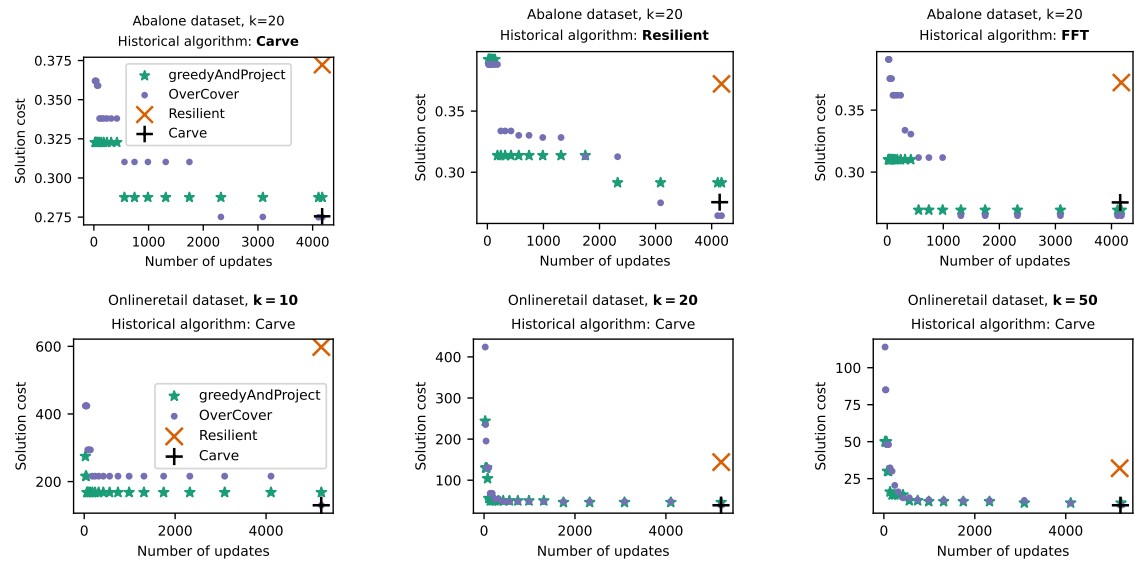

Figure 1: Comparison of our algorithm with the baselines for the first experimental setup.

in $\mathcal{H}$. Finally, we run ALG on $X$ with $\mathcal{H}$ as historical clustering. We record the score of the output clustering, for fixed $k$ and varying $b$.

**Setup 2: Evolutionary setting.** Consider a temporal dataset partitioned in $t$ slices $X = X_1 \cup \ldots \cup X_t$, where every data slice has equal size. We sequentially cluster the data: first, we run HIST on $X_1$ to obtain $C_1$. Then, for every $i \in \{2, \ldots, t\}$, we set $H = C_{i-1}$, and define $\ell_H(x)$ for every $x \in X_i$ to be the closest center from $x$ in $\mathcal{H}$, and we run ALG on $X_i$ with $\mathcal{H} = (H, \ell_H)$ as historical clustering. In the experiments below, we set $t = 20$. We record the score and the number of updates of the proposed solution on each of these data slices.

**Implementation choices.** In practice, we run both algorithms to get a solution with the aforementioned theoretical guarantees. Then, if there is budget remaining, we introduce a refinement phase, where we use the remaining budget to assign the furthest points to their closest center. Since this phase moves points to closer centers, it improves the objective value in practice, while preserving the theoretical guarantees.

Additionally, we modify OVERCOVER to remove its exponential running time in $k$. Instead of considering every possible choice for $H^*$, we construct it greedily: we remove as many historical centers as the budget permits, and define the remaining historical centers as our guess of $H^*$. This modified version of OVERCOVER runs in polynomial time, though at the cost of losing its approximation guarantee.

### 5.2 EMPIRICAL RESULTS

In this section, we evaluate the output quality of our algorithms and baselines, for the two setups described above. We discuss results on Figure 1 and Figure 2, while the complete results for the experiment is deferred to Appendix C.4. To avoid clutter in the figures, we only report results of RESILIENT and CARVE baselines, since FFT performs similarly to CARVE.

**Setup 1.** In Figure 1, we report results for the Abalone dataset, with HIST$\in \{$CARVE, FFT, RESILIENT$\}$ and $k = 20$, and results for the OnlineRetail dataset, with HIST$=$CARVE and $k \in \{10, 20, 50\}$.

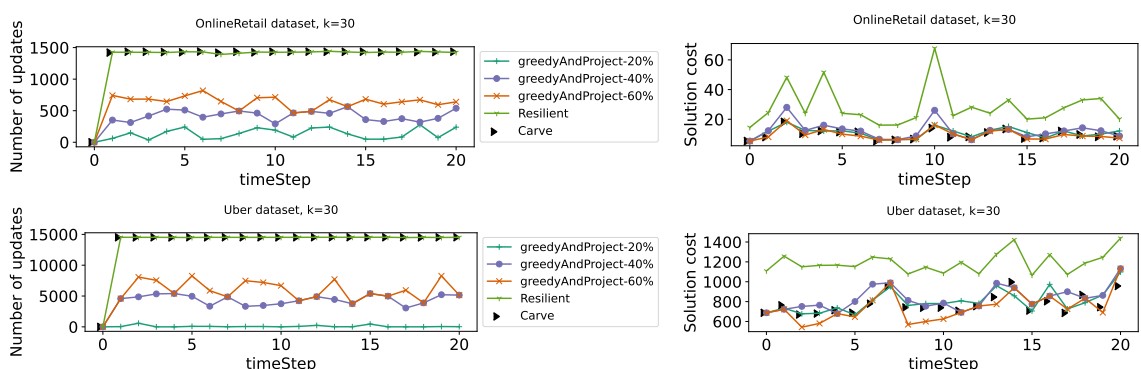

Figure 2: Comparison of our algorithm with the baselines for the second experimental setup.

First, we evaluate the number of updates of each algorithms. Note that only one point per baseline algorithm is reported as their input ignores the budget $b$. We notice that almost all points are reassigned for all the baselines, while our algorithms respect the budget.

Next, in term of solution cost, both GREEDYANDPROJECT and OVERCOVER outperform RESILIENT, even for small number of updates, and perform similarly to CARVE for larger values. For instance, for the OnlineRetail dataset, for $k = 50$, reassigning as little as $9\%$ of the points brings the cost to within $44\%$ of that of CARVE, while reassigning $49\%$ of the points further narrows the gap to just $22\%$.

Finally, OVERCOVER usually achieves better cost than GREEDYANDPROJECT for very high number of updates, since our implementation of OVERCOVER coincides with CARVE when $b = n$. However, GREEDY-ANDPROJECT usually outperforms OVERCOVER when less updates are permitted, suggesting better choices are made when choosing which historical center to preserve.

**Setup 2.** Results are reported in Figure 2 for GREEDYANDPROJECT, $k = 30$ and two datasets, with results for OVERCOVER and additional results for GREEDYANDPROJECT are reported in Appendix C.4. Here, GREEDYANDPROJECT$\alpha\%$ represents the problem instance where $b = \frac{\alpha}{100}n$. The observations made previously in the previous setup carries in this temporal setup: RESILIENT performs poorly due to each data slice being quite dissimilar with the preceding one, and CARVE usually performs better than GREEDY-ANDPROJECT but reassigns all the points.

Note that more budget does not guarantee better performance on each individual data slice, as the set of historical centers depends on the output of the algorithm on the previous timestep. However, we notice that more budget allows the algorithm to perform better on most slices: in the Uber dataset, GREEDYANDPROJECT$60\%$ outperforms GREEDYANDPROJECT$20\%$ $80\%$ of the time.

## 6 CONCLUSION AND FUTURE WORK

We introduced a novel family of problems LABEL-CONSISTENT $k$-CLUSTERING based on the notion of consistency, and we proposed two constant-factor approximation algorithms in the context of $k$-center clustering. Our main algorithm achieves a factor 3 approximation in polynomial time against a lower bound of 2, leaving open the question of closing this gap. It would also be valuable to study consistency applied to $k$-median or $k$-means, or to different data mining problems.

## ETHICS STATEMENT

This work is theoretical in nature and does not involve the collection or use of private, sensitive, or personally identifiable data. No studies involving human subjects were conducted. The methods developed focus on clustering from an algorithmic perspective, and we are not aware of any direct negative societal or ethical implications. While clustering techniques can in principle be applied in sensitive contexts, our contribution is methodological and abstract, and we leave considerations of application-specific impacts to future work.

## REPRODUCIBILITY STATEMENT

Complete proofs of all theorems presented in the main body of the paper are available in Appendix B. Moreover, all experimental results presented in the main body of the paper and in the appendix were produced using code publicly available in the supplementary material. The archive contains the code to download and preprocess the datasets, the implementation of all methods used in the experiments, as well as necessary code to reproduce the figures. All the plots presented here are, given enough time and compute resources, reproducible using one script.

## LLM USAGE

An LLM tool was used solely for light editing tasks such as grammar checking, typo correction, and other minor polishing. No LLM nor any other kind of generative AI was used for the development of our research ideas, literature review, implementation of our methods, and analysis of results.

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

## A  PSEUDOCODE FOR OVERCOVER

Pseudocode for OVERCOVER can be found in Algorithm 3, and pseudocode for CARVE in Algorithm 4.

---

**Algorithm 3** OVERCOVER    % 2-approximation in FPT time

---

**Input:** an instance $\mathcal{I} = ((X, d), k, \mathcal{H} = (H, \ell_H), b)$ of
  LABEL-CONSISTENT $k$-CENTER, optimal radius $r^*$
**Output:** consistent clustering $\mathcal{C} = (C, \ell_C)$
 1: guess $H^*$ the set of preserved historical center in an opti-
   mal solution
 2: $X_B \leftarrow X \setminus \cup_{h \in H^*} \text{Ball}(h, r^*)$
 3: $S' \leftarrow \text{CARVE}(X_B, 2r^*)$
 4: $C \leftarrow H^* \cup S'$
 5: **for** $x \in X$ **do**
 6:   **if** $\ell_H(x) \in H^*$ and $d(x, \ell_H(x)) \leq r^*$ **then**
 7:     $\ell_C(x) = \ell_H(x)$
 8:   **else** $\ell_C(x) \leftarrow \arg\min_{c \in C} d(x, c)$
 9: **return** $\mathcal{C} = (C, \ell_C)$

---

**Algorithm 4** CARVE

---

**Input:** Set $X'$ of points, $r \in \mathbb{R}_{\geq 0}$
**Output:** Set $S' \subseteq X'$
 1: $S' \leftarrow \emptyset$
 2: **while** $X' \neq \emptyset$ **do**
 3:   pick arbitrary point $s' \in X'$
 4:   add $s'$ to $S'$
 5:   $X' \leftarrow X' \setminus \text{Ball}(s', r)$
 6: **return** $S'$

---

## B  OMITTED PROOFS

### B.1  PROOF OF THEOREM 1

The proof follows from a simple polynomial time reduction from $k$-CENTER. Given an instance $\mathcal{J} = ((X, d), k)$ of $k$-CENTER, we construct an instance $\mathcal{I} = ((X, d), k, \mathcal{H} = (H, \ell_H), b)$ as follows: we pick an arbitrary set $H \subseteq X$ of size $k$, assign every point $x \in X$ to a closest center in $H$, i.e., $\ell_H(x) = \arg\min_{h \in H} d(x, h)$, and finally, set $b = |X|$. First, note that this is a polynomial time reduction in $|X|$. Next, we claim that the optimal cost of $\mathcal{I}$ is equal to the optimal cost of $\mathcal{J}$. Towards this, let $r^*_{\mathcal{I}}$ and $r^*_{\mathcal{J}}$ be the optimal costs of $\mathcal{I}$ and $\mathcal{J}$, respectively. It is easy to see that $r^*_{\mathcal{J}} \leq r^*_{\mathcal{I}}$ since any feasible solution to $\mathcal{I}$ is also a solution to $\mathcal{J}$. For the other direction, note that any solution to $\mathcal{J}$ is also a feasible solution to $\mathcal{I}$ since $b = |X|$. Hence, we have $r^*_{\mathcal{J}} = r^*_{\mathcal{I}}$. $i)$ and $ii)$ follows since, for any $\epsilon > 0$, $k$-CENTER is both **NP**-hard and **W**[**2**]-hard w.r.t. parameter $k$ to approximate to a factor $(2 - \epsilon)$ Vazirani (2001). Finally, for $iii)$, note that any solution to $\mathcal{J}$ reassigns at most $n \leq g(n) = g(b)$, as $b = n$, points, and hence is a feasible solution to $\mathcal{I}$, as well. $\qquad\square$

### B.2  PROOF OF THEOREM 2

OVERCOVER is our algorithm for LABEL-CONSISTENT $k$-CENTER, which is described in Algorithm 3. Let $\mathcal{C}^* = (C^*, \ell_{\mathcal{C}^*})$ be a fixed but unknown optimal solution to $\mathcal{I}$ with cost $r^*$. For the analysis, we assume that the algorithm has a correct guess of $r^*$ and $H^* = C^* \cap H$. We first claim that $\Delta(\mathcal{C}, \mathcal{H}) \leq b$. Towards this, let $X'_B = \{x \in X | \ell_H(x) \notin H^* \vee d(x, \ell_H(x)) > r^*\}$. Then, note that $\mathcal{C}^*$ has to reassign every $x \in X'_B$ to a center other than $\ell_H(x)$ in $C^*$ since $d(x, C^*) \leq r^*$. Since, $\Delta(\mathcal{C}^*, \mathcal{H}) \leq b$, we have $|X'_B| \leq b$. The claim follows since Algorithm 3 reassigns points only in $X'_B$ in the **for** loop.

Next, we claim that $|C| = |H^* \cup S| \leq k$. Towards this, let $\ell = |H^*| \leq k$, then we have to show $|S| \leq k - \ell$. Consider the set $X_B$ defined in Line 2, and assume $X_B \neq \emptyset$. Let $C^*_B = C^* \setminus H^*$. Then, note that $C^*_B \neq \emptyset$ since otherwise $\text{cost}_\infty(\mathcal{C}^*) > r^*$, a contradiction to $\mathcal{C}^*$. Furthermore, for every $x \in X_B$, it holds that $d(x, C^*_B) \leq r^*$. Therefore, $|S| \leq |C^*_B| \leq k - \ell$, since $(X_B, 2r^*)$ picks at most one point from each cluster of $C^*_B$ (Line 5).

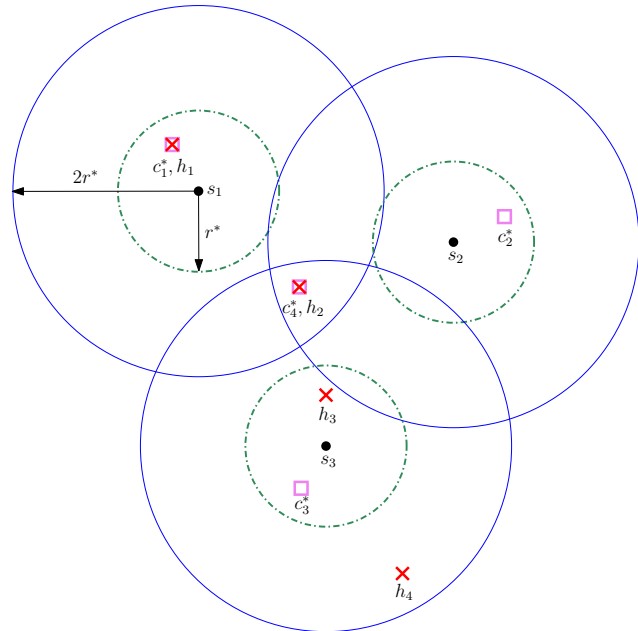

Figure 3: In the above figure, historical centers are represented by cross, optimal centers are represented by squares, and the elements of $S$ are represented as discs. For simplicity, we do not show remaining points of $X$. It can be seen that $H^* = \{h_1, h_2\}$, while $H_c^* = \{h_1\}$, and $H_f^* = \{h_2\}$. Also, $N_H(s_1) = \{h_1\}$, while $N_H(s_3) = \{h_3\}$. Furthermore, $\Gamma_{s_1}^* = \{h_1\} = H_c^*$, while all other $\Gamma^*$s are empty. Finally, suppose $w(h_4) > w(h_2)$, then the center set chosen by Algorithm 1 is $C = \{h_1, s_2, h_3, h_4\}$, and hence $w(C \cap H) = w(h_1) + w(h_3) + w(h_4) > w(h_1) + w(h_2) = w(C^* \cap H)$, and $\text{cost}_\infty(\mathcal{C}) \leq 3r^*$.

Finally, for the approximation guarantee, note that for $x \in X \setminus X_B$, we have $d(x, \ell_C(x)) = d(x, \ell_{C^*}(x)) \leq r^*$, while for $x \in X_B$, we have $d(x, \ell_C(x)) = d(x, C) \leq d(c, S) \leq 2r^*$. For the running time, it is easy to see that the algorithm runs in time $O(nk)$, since $\text{CARVE}(X_B, 2r^*)$ runs in time $nk$.

Guessing of $H^*$ can be done by considering every subset of $H$ as a candidate for $H^*$, which results in $2^k$ iterations of Algorithm 3. The algorithm also needs to verify whether or not the guesses of $H^*$ and $r^*$ are correct, which can be done by checking if the solution $\mathcal{C}$ satisfies $|C| \leq k$ and $\Delta(\mathcal{C}, \mathcal{H}) \leq b$. Finally, the algorithm returns a minimum-cost feasible solution. Therefore, the overall running time is bounded by $O(n^2 \log n + 2^k nk \log n)$. □

## B.3 PROOF OF LEMMA 1

Consider $s \in S$. Then, we know that $\Gamma_s^*$ are precisely the historical centers present in the optimal solution that are covered by $s$ (see Figure 3 for an illustration). On the other hand, let $\Gamma_s$ be the historical centers picked by the algorithm in $C_0' \cup C_1'$, i.e., $\Gamma_s = \hat{\Gamma}_s \cup \{\hat{s}\}$. Note that, for $s \neq s' \in S$, we have that $N_H(s) \cap N_H(s') = \emptyset$, since $d(s, s') > 2r^*$, and hence, $\hat{s} \neq \hat{s}'$ and $\Gamma_s^* \cap \Gamma_{s'}^* = \emptyset$, which in turn implies that $\Gamma_s \cap \Gamma_{s'} = \emptyset$. Therefore, we have that, the total number of historical centers in $C_0' \cup C_1'$ is lower bounded by $\sum_{s \in S_\gamma} |\Gamma_s| = \sum_{s \in S_\gamma} |\Gamma_s^*| = |H_c^*|$. Recall that, $H^*$ is the set of historical centers present in the optimal

solution $C^*$, and $w(H_c^*) + w(H_f^*) = w(H^*) \geq n - b$. We start by showing that the weight of the historical centers present in $C_0' \cup C_1'$ is at least the weight of the hidden historical centers, $w(H_c^*)$.

**Claim 3.** $w(H \cap (C_0' \cup C_1')) \geq w(H_c^*)$.

*Proof.* First note that $w(H \cap (C_0' \cup C_1')) \geq w(\cup_{s \in S_\gamma} \Gamma_s)$ since $H \cap (C_0' \cup C_1')) = \cup_{s \in S} \Gamma_s \supseteq \cup_{s \in S_\gamma} \Gamma_s$. Hence, it is sufficient to show that $w(\cup_{s \in S_\gamma} \Gamma_s) \geq w(\cup_{s \in S_\gamma} \Gamma_s^*) = w(H_c^*)$. Towards this, we show that for $s \in S_\gamma$, it holds that $w(\Gamma_s) \geq w(\Gamma_s^*)$, which implies that $w(\cup_{s \in S_\gamma} \Gamma_s) = \sum_{s \in S_\gamma} w(\Gamma_s) \geq \sum_{s \in S_\gamma} w(\Gamma_s^*) = w(\cup_{s \in S_\gamma} \Gamma_s^*) = w(H_c^*)$, as required.

Consider the **for** loop execution at Line 2 for $s \in S_\gamma$, and consider $\Gamma_s = \{\hat{s}\} \cup \hat{\Gamma}_s$ and let $\Gamma_s^* = (h_s^1, h_s^2, \ldots h_s^{|\Gamma_s^*|})$ be ordered in non-increasing order of the weights of the historical optimal centers. Then, note that $w(\hat{s}) \geq w(h_s^1)$ as the set $\Gamma_s^*$ is available for the algorithm to be picked for $\hat{s}$. Now consider the case when $\hat{s} \notin \Gamma_s^*$. Since all the centers in $\Gamma_s^*$ are available to be picked by the algorithm in Line 7, we have that $w(\hat{\Gamma}_s) \geq w(\{h_s^1, \ldots, h_s^{|\Gamma_s^*|-1}\})$, and thus, $w(\Gamma_s) = w(\hat{s}) + w(\hat{\Gamma}_s) \geq w(h_s^1) + w(\{h_s^1, \ldots, h_s^{|\Gamma_s^*|-1}\}) \geq w(\Gamma_s^*)$, as required. Now consider the case when $\hat{s} \in \Gamma_s^*$, and therefore $\hat{s} = h_s^1$ since it is the highest weight historical center in $\Gamma_s^*$. This means that, the centers $\{h_s^2, \ldots, h_s^{|\Gamma_s^*|}\}$ are available for the algorithm to pick in Line 7. Therefore, $w(\Gamma_s) \geq w(h_s^1) + w(\{h_s^2, \ldots, h_s^{|\Gamma_s^*|}\} = w(\Gamma_s^*)$, as required. $\square$

Next, we show that $|C'| \leq k$. Towards this, suppose $|S| = \ell$, and consider the optimal clusters $\{\pi_i^*\}_{i \in [k]}$. We say that a optimal cluster $\pi_i^*$ is *hit* by $S$ if $\exists s \in S$ such that $s$ belongs to cluster $\pi_i^*$. Since, the size of $S$ is $\ell$, and elements of $S$ belong to different clusters of the optimal solution, we have that $S$ hits exactly $\ell$ optimal clusters. Therefore, the number of *unhit* optimal clusters is $k - \ell$. On the other hand, $|C_1' \cup C_2'| = |C_1'| + |C_2'| = \sum_{s \in S_\gamma} (|\Gamma_s^*| - 1) + |H_f^*|$ is upper bounded by the number of unhit optimal clusters, while $|C_0'| = |S| = \ell$. Hence, we have $|C'| = |C_0'| + |C_1'| + |C_2'| \leq k$, as desired. $\square$

Now, we show that $w(H \cap C') \geq n - b$. Consider the hidden historical optimal centers, $H_f^*$, to $S$. Since, it holds that $H_f^* \cap (\cup_{s \in S} N_H(s)) = \emptyset$, we have that $H_f^* \cap (C_0' \cup C_1') = H_f^* \cap (\cup_{s \in S} \Gamma_s) = \emptyset$. Thus, all the centers in $H_f^*$ are available to be picked by the algorithm in Line 9 for $C_2'$. This means $w(C_2') \geq w(H_f^*)$, and hence

$$
\begin{aligned}
w(H \cap C') &= w(H \cap (C_0' \cup C_1')) + w(C_2') \\
&\geq w(H_c^*) + w(H_f^*) \\
&= w(H^*) \\
&\geq n - b,
\end{aligned}
$$

where the first equality follows since the sets $C_0', C_1', C_2'$ are pairwise disjoint and the fact that $H \cap C_2' = C_2'$, and the first inequality follows due to the above claim. Now, consider the **for** loop in Line 11 that assigns points to a center in $C$. Note that for every $x \in X$, it assigns $x$ to $\ell_H(x)$ if $\ell_H(x) \in C'$ and $d(x, \ell_H(x)) \leq r^*$. Therefore, for every historical center $h \in C' \cap \mathcal{H}$, the number of points assigned to $h$ is at least $w(h)$. Hence, the number of points reassigned by Algorithm 2 is at least $w(H \cap C') \geq n - b$, finishing the proof of the lemma. $\square$

### B.4 Proof of Lemma 2

Consider the center sets $C = C_0 \cup C_1$ and $C' = C_0' \cup C_1' \cup C_2'$ returned by Algorithm 1 and Algorithm 2, respectively. Then, note that $C_0 = C_0'$. Consider Line 7 of Algorithm 1 that constructs $C_1$. Since $C_0 = C_0'$, we have that all the historical centers in the set $C_1' \cup C_2'$ picked by Algorithm 2 are available for Algorithm 1

as candidate centers for $C_1$ in Line 7. Furthermore, $|C_1' \cup C_2'| \leq (k - |S|) = (k - |C_0|)$. Therefore, we have that $w(C_1) \geq w(H \cap (C_1' \cup C_2'))$, and hence

$$
\begin{aligned}
w(H \cap C) &= w(H \cap (C_0 \cup C_1)) \\
&= w(\mathcal{H} \cap C_0) + w(C_1) \\
&\geq w(\mathcal{H} \cap C_0') + w(H \cap (C_1' \cup C_2')) \\
&\geq w(\mathcal{H} \cap C') \\
&\geq n - b.
\end{aligned}
$$

The lemma follows since in the assignment routine (**for** loop in Line 9) of Algorithm 1, the number of points assigned to $h \in C \cap \mathcal{H}$ is at least $w(h)$. □

### B.5   PROOF OF THEOREM 3

Algorithm 1 gives a 3-approximation for LABEL-CONSISTENT $k$-CENTER by Theorem 4, but it requires the optimal radius $r^*$ as an additional input. A straightforward approach is to try all $n^2$ pairwise distances as guesses for $r^*$ and return the minimum-cost feasible solution. However, we can accelerate this by performing a binary search over these $n^2$ candidate values, yielding a total running time of $O(n^2 \log n + nk \log n)$, as desired. □

## C   ADDITIONAL EXPERIMENTAL RESULTS

### C.1   DATASETS DESCRIPTION

The *Abalone* dataset Nash et al. (1994) dataset records different measurements on abalones. We keep all features, except the sex and the number of rings. The dataset contains 7 features and 4176 entries.

The *Electric Consumption* dataset Hebrail and Berard (2006) dataset contains 207.5 million power measurement in Sceaux, France, between Dec. 2006 and Nov. 2010. We retain measurements occurred during the first 20 months, for a total of 439820 entries, and use 7 numerical features.

The *OnlineRetail* dataset Chen (2015) records 54.2 million transactions of online retail, occurring from Dec. 2010 to Dec. 2011. We retain the two numerical features, and transactions occurring between Jan. 1, 2011 and Jan. 20, 2011, for a total of 28300 transactions.

The *Twitter* dataset Helwig et al. (2015) record seven days of geo-tagged Tweet data from the United States, sent between Jan 12, 2013 and Jan 18, 2013. We keep tweets sent during the first 20 hours, for a total of 289860 tweets.

The *Uber* dataset[4] contains around 18.8 million Uber pickups in New York City from April to June 2015. We retain pickups between Jun 1, 2014 and Jun 20, 2014, for a total of 290200 pickups.

In both the Twitter dataset and the Uber dataset, we convert the angular Geo location to Cartesian coordinates, for a total of 3 features.

### C.2   EXPERIMENTS ON $\varepsilon$-CLOSE POINT SETS

This additional experiments was conducted to study the empirical performance of our algorithm on instances where when data are affected by some noisy perturbation, which are instances where RESILIENT is designed to perform well.

---

[4]https://www.kaggle.com/datasets/fivethirtyeight/uber-pickups-in-new-york-city

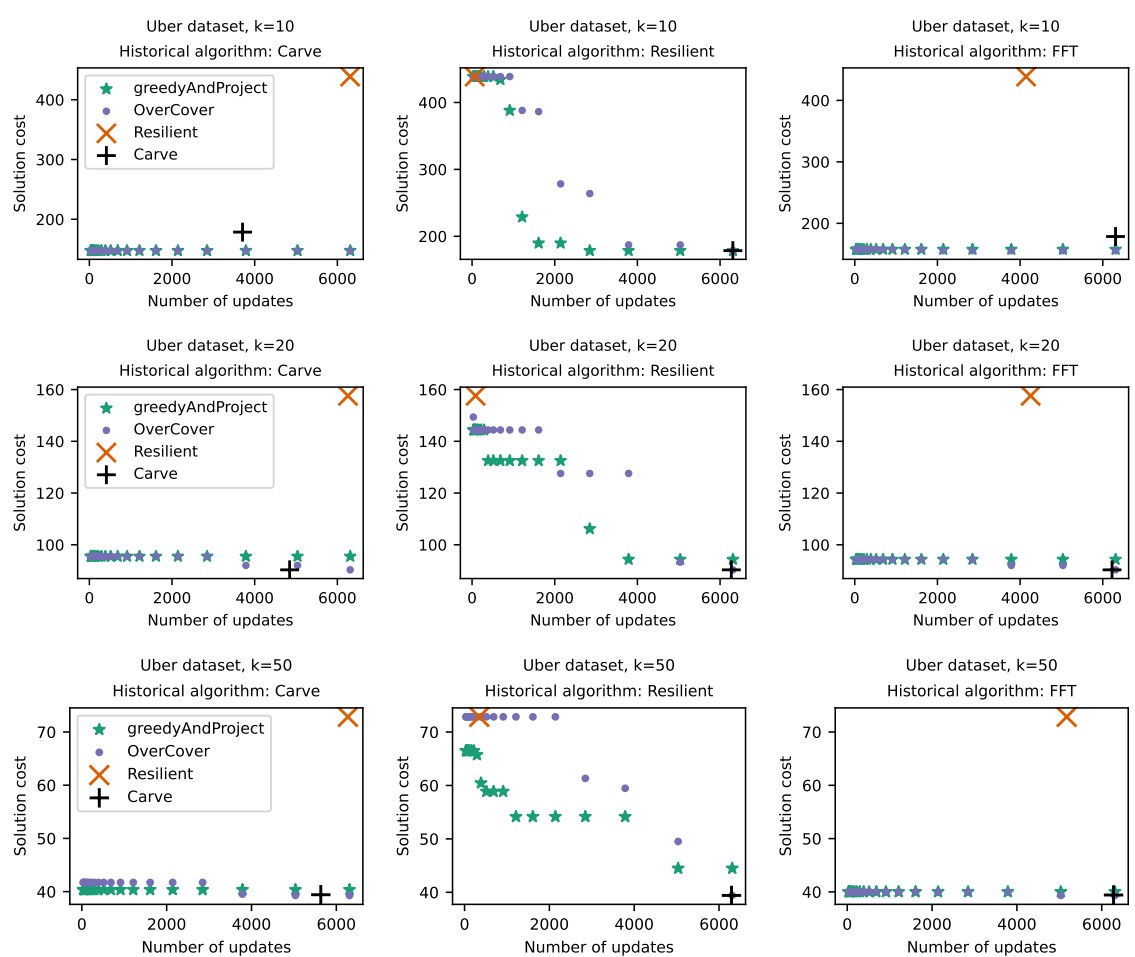

Figure 4: Comparison of our algorithm with the baselines for the third experimental setup.

We use the Uber dataset, and follow the same pre-processing method as by Ahmadian et al. (2024): we consider the pickup locations of the first two days of June 2014, and convert the angular Geo location to Cartesian coordinates. Then, we find a minimum-weight perfect matching between points of the first and second day, and remove unmatched locations as well as matched locations located more than 1 km apart. Finally, we denote by $X'$ the dataset for the first day, and $X$ the dataset for the second day. We run HIST on $X'$ to obtain $\mathcal{H}$, and convert the historical centers $H$ to their matched points in $X$. Then, we run ALG on $X$ with $\mathcal{H}$ as historical clustering.

**Empirical results.** Results for this setup are reported in Figure 4. Since the points in the first and second day datasets are similar, RESILIENT manages to find similar clusters, provided that RESILIENT or FFT is used as historical clustering. For example, RESILIENT achieves as little as $2\%$ reassigned points when using RESILIENT historical centers with $k = 30$, and $80\%$ when using FFT as historical clustering. This cross-compatibility exists because RESILIENT uses FFT as a subroutine to find some centers. Note that

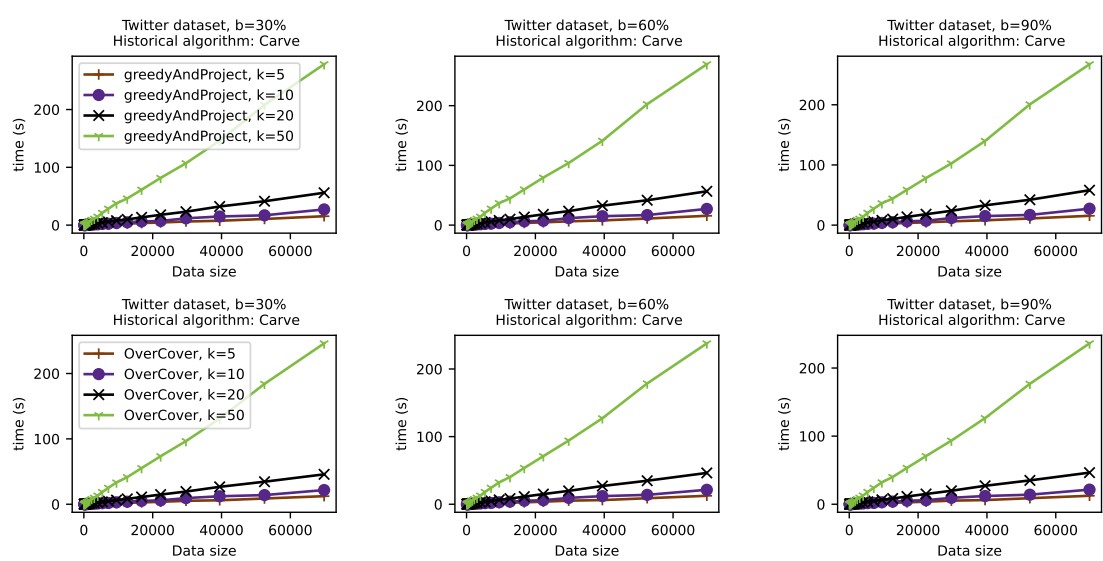

Figure 5: Scalability of our methods

the consistency is lost when the historical clustering comes from a different clustering algorithm, such as CARVE. Note that both FFT and CARVE achieve lower clustering cost than RESILIENT.

Meanwhile, with both OVERCOVER and GREEDYANDPROJECT, we can perform similarly to RESILIENT and CARVE, on both the number of reassigned points and clustering score, with an additional control of in-between number of updates.

### C.3 RUNNING TIME ANALYSIS

We compare running time of GREEDYANDPROJECT and OVERCOVER for different $k$ values, on the Twitter dataset, and report the running times on Figure 5. First, we can notice that the running time seems independent of the number of updates $b$. Furthermore, for fixed value of $k$, the running time scales of both algorithm scales well with respect to the size of the dataset.

### C.4 ADDITIONAL RESULTS FOR EXPERIMENTAL SETUP 1 AND 2

We provide additional plots for the experimental setups described in Section 5.1. For setup 1, this includes all combinations of historical clustering and datasets, for $k = 10$ in Figure 6, $k = 20$ in Figure 7 and $k = 50$ in Figure 8. For setup 2, this includes plots for the Electricity and Twitter dataset for GREEDYANDPROJECT in Figure 9, and plots for OVERCOVER in Figure 10.

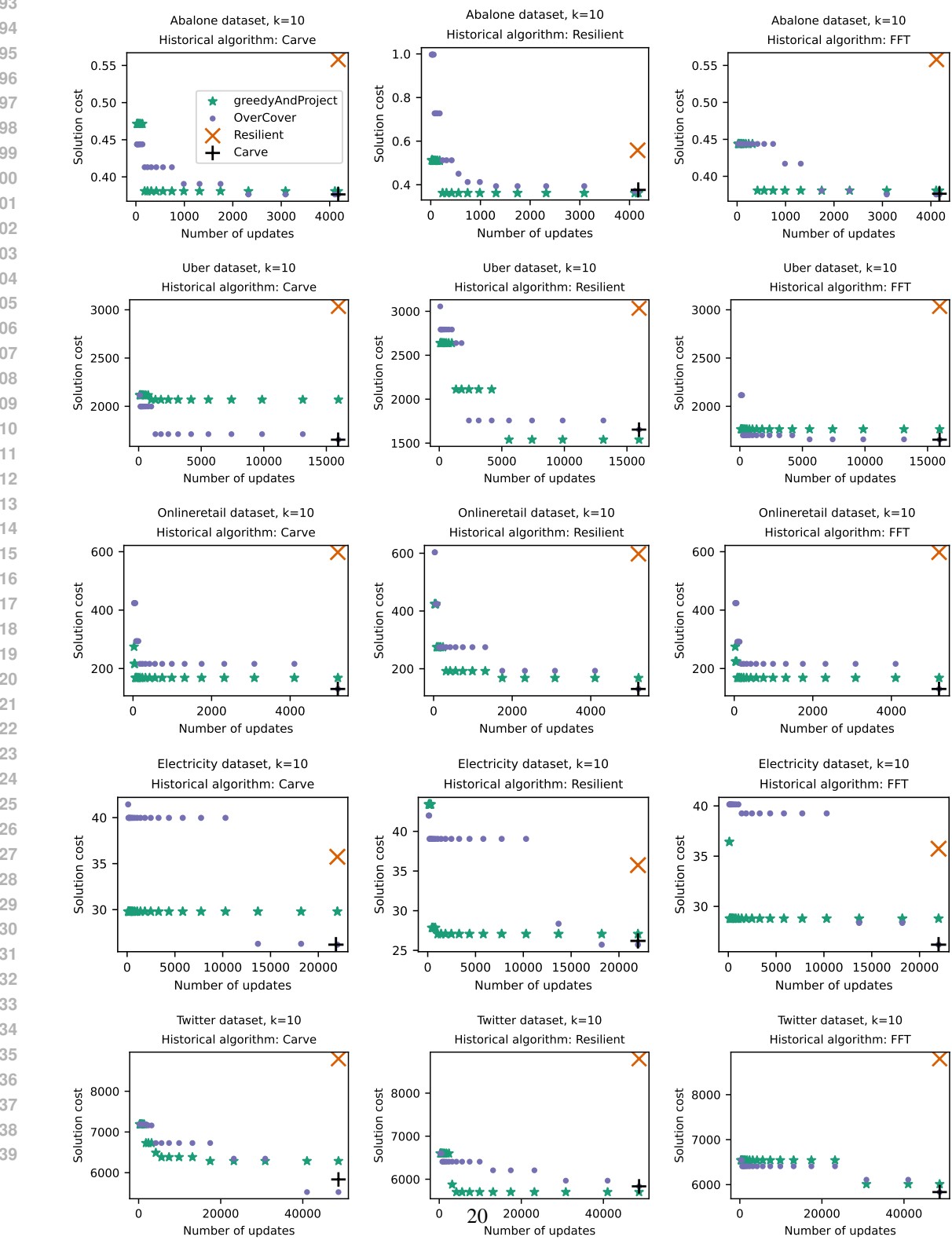

Figure 6: Extra plots for the first experimental setup, $k = 10$.

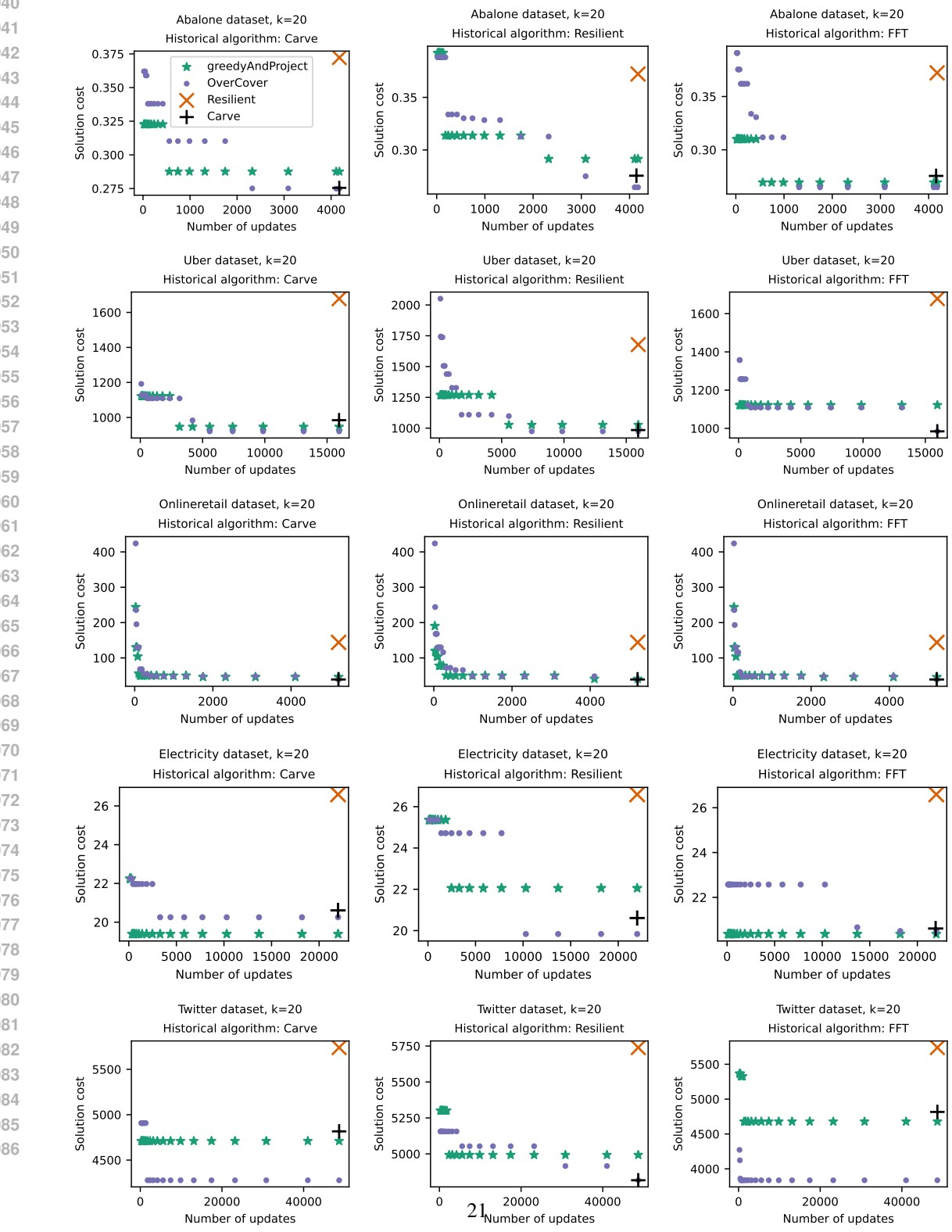

Figure 7: Extra plots for the first experimental setup, $k = 20$.

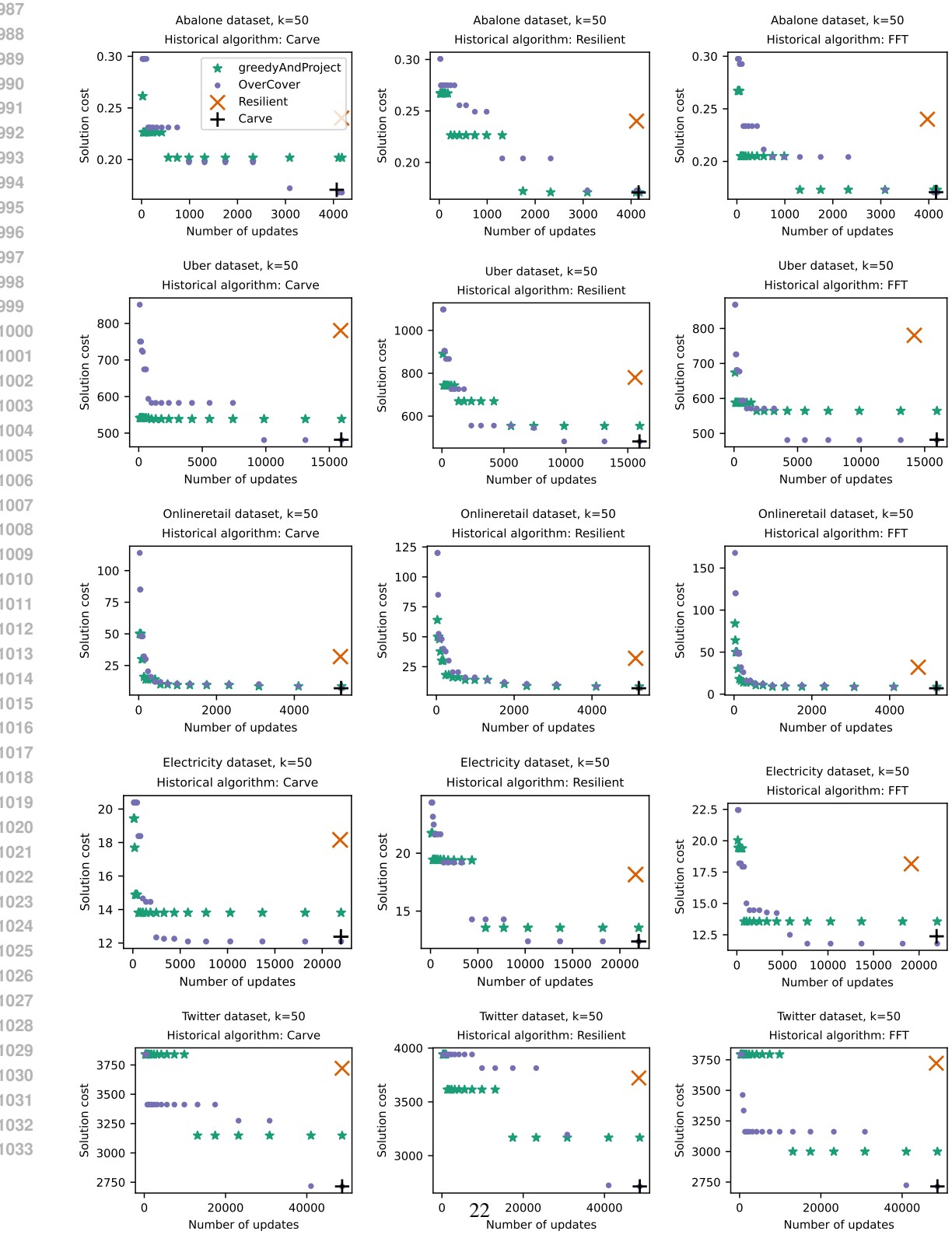

Figure 8: Extra plots for the first experimental setup, $k = 50$.

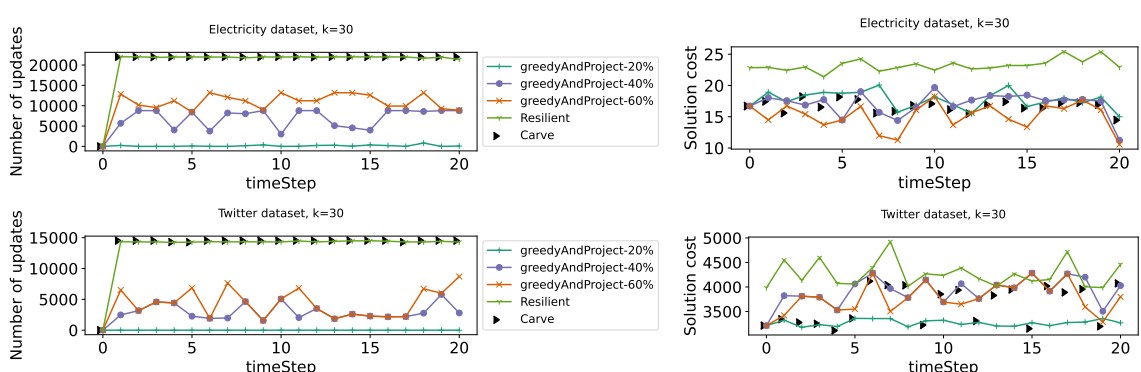

Figure 9: Extra plots for the second experimental setup, for GREEDYANDPROJECT.

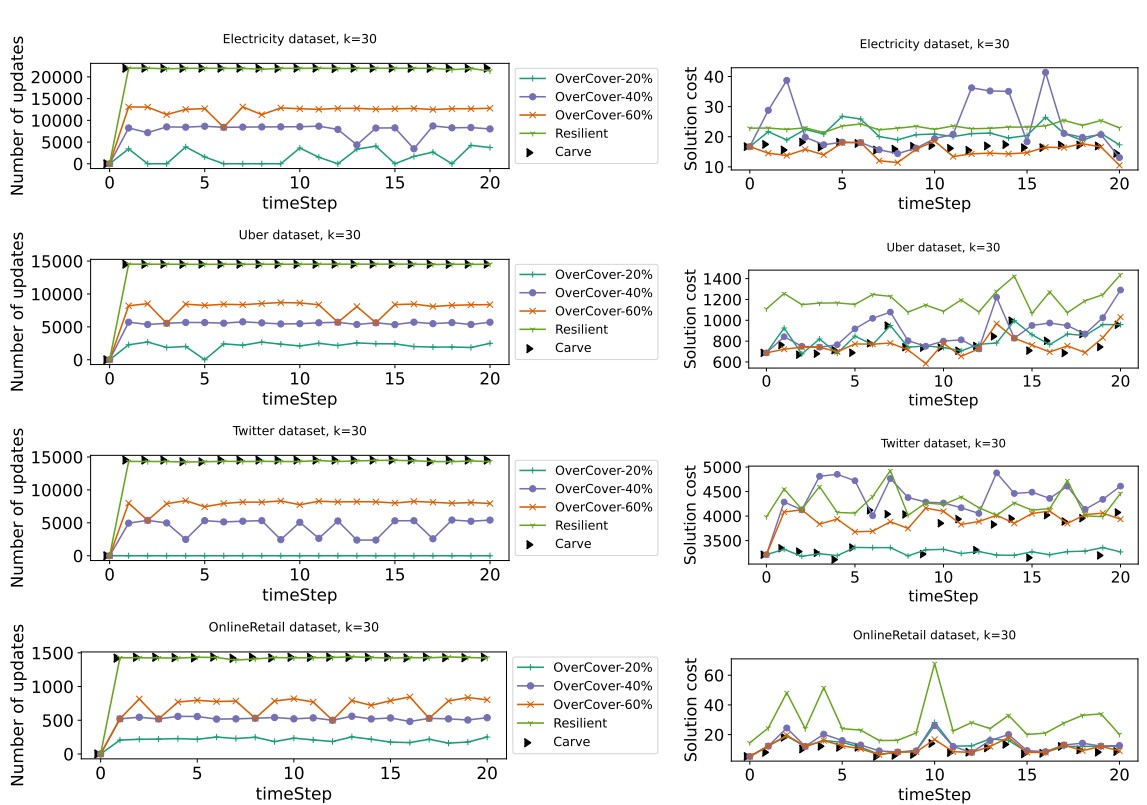

Figure 10: Extra plots for the second experimental setup, for OVERCOVER.