# OpenReview forum: "Label-consistent clustering for evolving data"
_ICLR.cc/2026/Conference — Submitted to ICLR 2026_

### Official Review · Reviewer_KFw2 · 2025-10-27

**Soundness:** 3
**Presentation:** 2
**Contribution:** 2
**Rating:** 4
**Confidence:** 4

**Summary:**

The paper introduces a new problem formulation in the context of k-center (or in general center-based) clustering, named label-consistent clustering. In this problem, which is strongly related to the problem of minimizing recourse, one is given a previous solution (i.e. a set of centers) to the pointset at hand, as is tasked with finding a new solution that minimizes the objective function while introducing at most b changes to the assignment of points to their centers. The paper presents two constant-factor-approximation algorithms, provides a theoretical analysis of the algorithms, and provides some experimental evaluation of them.

**Strengths:**

- The definition of label-consistent clustering, an alternative definition of recourse, makes sense and could have practical implications. Therefore, the introduction of a new problem seems justified.
- The two proposed algorithms seem to be a convincing contribution, as they provide a tradeoff between approximation quality and runtime.
- The experimental evaluation in an interesting addition to the theoretical analysis.

**Weaknesses:**

- The definition of the problem is slightly unclear at times. The paper supposes an evolving pointset, akin to the fully dynamic setting. The task tries to minimize changes in the point assignments in the pointset $X_t$ given a solution $\mathcal H$ of the pointset $X_{t-1}$ at time $t-1$ (e.g. see lines 50-57). However, in Definition 3, it seems like $\mathcal H$ is already a solution for $X_t$. This might create some problems, as for example some points that were centers in $X_{t-1}$ might not be in in $X_t$. Therefore, $\mathcal H$ might not be a subset of $X$. Would the definition still be valid? And would the algorithms still work?
- Theorem 1 is hardly a contribution, as it follows directly from the hardness of k-Center.
- Issues with the presentation of the algorithms:
    - In Section 4.2, the authors should explain that they take all distances, sort them and binary search the correct value for $ r^* $, as right now, for anyone outside the clustering community, it would be really hard to understand how the algorithm would have access to $ r^* $ without reading the appendix.
    - In line 7 of Algorithm 1, points are taken from $H \setminus C$, but $C$ should be empty. Should it be $H \setminus C_0$?
    - At first sight, it seems like $b$ is never used in the algorithm, which is puzzling. In fact, $b$ seems to be used to check if the solution is feasible when looking for the correct value of $r^*$. This should be explained clearly.
- The paper has a superficial literature review, and there seems to me some missing related work:
    - The paper only briefly discusses the literature on consistent clustering, which tries to control recourse. [Lattanzi and Vassilvitskii, 2017], [Lacki et al, 2024] and [Forster and Skarlatos, 2025] are cited, but not discussed in detail. The authors should clearly explain the differences between previous work and label-consistent clustering. [1] also deals with recourse, but was not cited.
    - Consistent or low-recourse algorithms have been proposed for problems other than clustering, but have not been discussed. E.g. see [2], [3] and references therein.
    - The paper does not discuss dynamic k-center algorithms (e.g. [4], [5], [6]), even though this framework seems to be the main motivation for this paper.
- There is no indication whether the gap between the lower bound of 2 and the upper bound of 3 can be closed with an algorithm that is polynomial time (i.e. not FPT) or not. Such a lower bound or a 2-approximation algorithm would have made the contribution much stronger. This is of course a *minor* weakness, I understand that achieving this might be really hard.

----
References:

- [1] Fichtenberger et al. Consistent k-Clustering for General Metrics. SODA 2021.
- [2] Bhattacharya et al. Chasing Positive Bodies. FOCS 2023.
- [3] Bernstein et al. Online Bipartite Matching with Amortized O(log2n) Replacements. J. ACM 2019
- [4] Chan et al. Fully Dynamic k-Center Clustering. WWW 2018
- [5] Pellizzoni et al. Fully Dynamic Clustering and Diversity Maximization in Doubling Metrics. WADS 2023
- [6] Bateni et al. Optimal Fully Dynamic k-Center Clustering for Adaptive and Oblivious Adversaries. SODA 2023.

**Questions:**

- the choice to prove correctness of Algorithm 1 via Algorithm 2 seems odd. Do you think there is a simpler proof?

----

Remark:

Overall, I think the paper has some clarity issues, and the algorithmic contribution is not major. However, I am willing to raise my score if the clarity is improved.

---

> ### Author Response · Authors · 2025-11-21
>
> We thank the reviewer for the thorough review and constructive comments.
>
> **Regarding W1:** This point is addressed in the common answer above.
>
> **Regarding W2:** Theorem 1 is included for clarity of exposition, and we do not claim it as our contribution in the paper.
>
> **Regarding W3:** We will correct the typos in the revised  manuscript, and provide additional explanations on  how to find $r^*$ (as discussed in the appendix), as well as clarify how the budget is used in Algorithm $1$. Additionally, line $7$ of Algorithm $1$ should read $H \setminus C_0$, as mentioned by the reviewer.
>
> **Regarding W4:** While we already discuss how our clustering distance function varies from the one proposed in consistent clustering (see lines 257--272 of the manuscript), we plan to add the additional comparisons to the related work section, as per the suggestion. As explained in the common answer, our setting differs fundamentally from the dynamic setting (even with low recourse). In dynamic settings, the algorithm must competes with **all solutions**, whereas in our setting, it only  competes only with the solutions that satisfy the hard budget constraint. Hence, our setting is closer in spirit to resilient or consistent clustering, as already noted in the paper.
>
>
> **Regarding W5:** Regarding the gap between the upper and lower bounds for polynomial-time algorithms, we will propose this as an open question in the revised manuscript. We would like to point out such gaps are not unique to our problem. For instance, similar gaps exist between the standard and constrained versions of $k$-center, such as capacitated $k$-center, fair-range $k$-center, and it is an open question to close them.
>
> **Regading Q1:** We respectfully disagree with the reviewer’s opinion that proving the correctness of an algorithm using a simpler algorithm is unusual. In fact, Algorithm 2 highlights the key ideas of Algorithm 1 while abstracting away non-essential steps, and we believe this leads to clean and conceptually transparent analysis.
> Such techniques are common in algorithm analysis, particularly when the main algorithm is complex and the analysis algorithm is simpler. The simplest example is analyzing the greedy 2-approximation algorithm for vertex cover using maximum matching [1] and (Section 1.2 of [6]). Further examples include:
> dual-fitting for greedy algorithm for set-cover (Chapter 13 of [1,3]),
> improved analysis of greedy for Vertex-cover using dual fittings (Chapter 1.6 of [2]),
> primal-dual for Dijkstra’s algorithm for shortest path (Section 7.3 of [2], [3]),
> using duality to analyze (and design) approximation algorithms (Chapter 52 of [4]), and using the pricing method to analyze greedy (Chapter 11.4 of [5]).
>
>
> [1] Approximation Algorithms, Vazirani
>
> [2] The design of approximation algorithms, Williamson and Shmoys
>
> [3] https://www.cs.jhu.edu/~mdinitz/classes/ApproxAlgorithms/Spring2019/Lectures/lecture20.pdf
>
> [4] https://www.cs.umd.edu/users/samir/2012lecnotes.pdf
>
> [5] Algorithm Design, Kleinberg and Tardos
>
> [6] https://tandy.cs.illinois.edu/dartmouth-cs-approx.pdf

---

### Official Review · Reviewer_hJwX · 2025-10-31

**Soundness:** 3
**Presentation:** 4
**Contribution:** 2
**Rating:** 4
**Confidence:** 4

**Summary:**

This paper introduces and studies the problem of updating an existing historical $k$-center clustering solution $H$ when the dataset gets updated, under the constraint that only $b$ points are allowed to change their centers. They present a $2$ approximation (to the best possible $k$-center clustering under these constraints) running in FPT time in the number of centers, and a $3$ approximation running in polynomial time. They also extend their algorithms to generally work with evolving data by using the output of the previous time step as the historical solution to the current time step. Both their algorithms start by assuming they know the radius of an optimal label-consistent clustering, $r^*$, since they can obtain it by binary searching for the best value.

Their first contribution is an algorithm OverCover that gives a $2$ approximation. The algorithm works by guessing the set of centers in $H$ that do not change in a new optimal solution, called $H^*$, then uses the classic algorithm of Hochbaum and Shmoys on the remaining unclustered data to obtain a $2$ approximation. The guessing of centers causes the blow up in the terms that depend on $k$.

Their second contribution is a $3$-approximation algorithm that runs in polynomial time. In the first phase, the algorithm runs the classic algorithm above to obtain a set of centers. In the next phase, each center in this set is swapped with a center from its own cluster that is also a historical center. The historical center with the highest weight is chosen, which is the number of points currently within an $r^*$ radius of the center that also used to belong to its cluster in $H$. Finally, historical centers are added in decreasing order of their weights until the current solution reaches $k$ centers.

They also extend existing lower bounds to show that label-consistent $k$-center is NP hard to approximate within a less than 2 factor, and also W[2]-hard to approximate to the same factor when parameterized in the number of centers.

Finally, they evaluate slightly modified versions of their algorithms against the classical one by Hochbaum and Shmoys, the FFT algorithm that iteratively searches for the farthest point to explore, and the algorithm by Ahmadian et al. for resilient clustering.

Summary recommendation: I would weakly recommend rejecting the paper due to the above-mentioned doubts. Specifically, I do not immediately get the motivation for defining the model in this way as well, which, if the authors could talk about in their rebuttal, would be useful for situating their contributions more in context. The theoretical part and the experimental part also have coherent messages separately, but I could not see if the algorithms implemented had any of the guarantees that were proven earlier. A discussion about this (if it does satisfy some of the guarantees) would be useful.

**Strengths:**

- The paper presents algorithms that are theoretically neat and simple and give provable guarantees.
- The paper is generally written well and easy to read and follow.
- The experiments show that (a modified version of) their algorithms are better than the baselines.

**Weaknesses:**

- As I can see, there are two ways of asking for recourse bounds in the setting of evolving data: One is the method taken here, which is to consider the case where there already exists a historical solution (which does not arise from the algorithm itself). Another, which I've personally seen more of, is to ask that the algorithm maintains a good approximation to the overall best solution, but with low recourse. In the latter setting, the guarantee is measured with respect to all clusterings instead of just the ones within swap distance $b$ of the current clustering. Is there a reason for choosing the former (which also gives guarantees with respect to really bad historical clusterings) here? Are they equivalent in some way? Is there a reason for considering the setting where the algorithm might get bad historical solutions that it did not come up with but was given to it as a warm start? Are there bounds that say that any algorithm that maintains a close-to-optimal solution must have high recourse necessarily?
- The first algorithm requires iterating over all possible $H^*$s which is very costly and not part of the implemented version of the algorithm, which makes it practically infeasible. One can implement it without the iteration, but then does it have any theoretical guarantees?

**Questions:**

Typos:
line 045: re-assinged -> re-assigned
line 183: that present -> that are present
line 215: "a set S of k centers such that ... |S| \le k"
line 246: should the call to CARVE be r^* instead of 2r^*?
line 307: deferred in -> deferred to
line 322: algorithms -> algorithm
line 701: instanes -> instances
line 751: due because -> because?

---

> ### Author Response · Authors · 2025-11-21
>
> We thank the reviewer for the thorough review and constructive comments. We also thank the reviewer for pointing out the typos, which we will be corrected in the revised  manuscript.
>
> **Guarantee algorithms used in experiments:** We would like to emphasize that we have implemented our main algorithm ($3$-approximation GreedyAndProject), and the corresponding experimental results are included in the paper. We note that the practical implementation of GreedyAndProject includes one additional phase (referenced in lines 365--366), but this step does not affect the algorithm's theoretical guarantees. We will make this clarification more prominent in the revised version. However, as you pointed out in W2, OverCover is unusable in practice, due to its exponential running time in $k$. Thus, OverCover was modified to have a reasonable running time, though at the cost of its approximation guarantee. We will add this important clarification in the revised manuscript.
>
> **Regarding W1:** We refer to the common response on the comparison to dynamic setting. We plan to clarify the points raised in the common answer in the revised version.
>
> Regarding your questions about the problem setting and problem formulation we propose, our choices are motivated by scenarios where the historical clustering is **fixed and immutable**, that is, it has already been presented to the user (e.g., a set of clients assigned to services, which has already been published) and cannot be retracted.
> In such a case, even if the historical clustering is imperfect, in order to avoid disruptions we may want to require that the new clustering stays consistent to the historical clustering.
> In other words, if consistency is an important requirement, the best approach is to minimize the clustering error **given** the data updates and **given** the historical clustering (even if the latter is imperfect) while satisfying the consistency constraints.
>
> We do not claim that our approach is the only way to handle evolving data, but it is one possible way that is **complementary** to existing approaches and, in our opinion, is better suited for certain applications, such as those involving fixed and immutable historical clustering as described above.
>
> In other cases where we can warm-start the historical clustering, we can start  with a good approximate clustering as a historical clustering, ensuring near-optimality. If the clustering degrades as data evolves, we can re-cluster the points (ignoring $b$) and restart the evolution.
>
> We will include this explanation in the revised manuscript.
>
> **Regarding W2:** We agree that iterating over all $H^\*$s is computationally expensive, and we currently do not know a more efficient way to find such sets. Our main motivation for this algorithm, besides its theoretical interest, is to serve as a warm-up for our main algorithm, which is significantly more efficient and loses only a small additional factor in approximation.
> At a high level, the algorithm attempts to select $H^\*$ though it may include other historical centers in the solution. The key property is that it opens at least $|H^\*|$ historical centers, respecting the budget constraint while incurring only a slight extra approximation factor. We view these two algorithms as representing a tradeoff between approximation quality and running time. While there may exist efficient methods to find $H^\*$, we are currently unaware of any, and hence this is also one of the open problems in our work.

---

### Official Review · Reviewer_2CHi · 2025-11-01

**Soundness:** 3
**Presentation:** 3
**Contribution:** 2
**Rating:** 6
**Confidence:** 3

**Summary:**

The paper studies the Label-consistent $k$-clustering problem, which aims to find a new clustering on an evolving dataset while keeping the labeling consistent with a given historical clustering. Specifically, the objective is to minimize the clustering cost subject to the constraint that the number of points whose cluster labels change does not exceed a given budget $b$. The authors propose a tight FPT 2-approximation and a polynomial-time 3-approximation algorithms, and provide experiments on both synthetic and real datasets to demonstrate the effectiveness of their approach.

**Strengths:**

1. The formulation captures an important setting in incremental or evolving data analysis, where stability across time steps is crucial for interpretability and system reliability.
2. The paper offers both provable guarantees and empirical validation, which strengthens its technical completeness.

**Weaknesses:**

1. While the introduction and experiments emphasize the incremental or evolving nature of the data (e.g., extending a historical clustering to a new dataset at time $t$), the formal definition (Definition 3) treats the instance as a static dataset $𝑋$. It would improve clarity if the authors explicitly defined how the dataset evolves, e.g., distinguishing between existing and newly arrived points, and how the label consistency constraint applies in that setting.

2. In the problem definition, the threshold $b$ bounds the allowed number of label changes, yet it is not clearly stated whether it is an input parameter. Since the algorithms and experiments rely on $b$, the paper would benefit from a short clarification of how $b$ is chosen or interpreted.

3. The paper mainly discusses data additions (incremental updates), but not the case where some previous data points are removed. A short remark on how the approach could be adapted to handle such deletions would make the formulation more complete.

**Questions:**

1. In the formal definition, the dataset $X$ appears static. How should readers interpret the **incremental** setting mentioned in the motivation. Does $X$ consist of both old and new data, or only newly added points? Could you make this distinction explicit to better reflect real incremental scenarios?

2. How would the proposed approach handle data deletions—i.e., when some previously clustered points are removed from the dataset? Would the label-consistency constraint or the budget $b$ need adjustment?

3. Do you think the same framework or algorithmic idea could be extended to other objective functions such as $k$-median or $k$-means? Since the definition of label consistency is independent of the specific cost function, a short discussion on this potential generalization would be valuable.

---

> ### Author Response · Authors · 2025-11-21
>
> We thank the reviewer for the thorough review and constructive comments.
>
> Regarding W1, W3, Q1 and Q2 (handling new data), see the common answer.
>
> **Regarding W2:** We will clarify in the revised version that budget $b$ is an input parameter of the problem and controls the degree of *label consistency* in the resulting clustering. In practice, we chose to set $b$ as a proportion of the total amount of point (i.e. $20$%, $40$% or $60$%), to evaluate the portion of points to reallocate. In the experiments, we explore those different values of $b$ to study the impact of the label consistency on the resulting clustering.
>
> **Regarding Q3:** A generalization to label-consistent $k$-means or $k$-median would indeed be highly valuable. However, the algorithms developed in this paper are specifically tailored to the $k$-center objective, and we do not expect them to extend to $k$-means or $k$-median in a straightforward manner. We have therefore listed these generalizations as open problems in Section 6.

---

### Official Review · Reviewer_dpc9 · 2025-11-03

**Soundness:** 2
**Presentation:** 2
**Contribution:** 2
**Rating:** 2
**Confidence:** 5

**Summary:**

This paper proposes the "Label-Consistent $k$-Clustering" problem, aiming to compute a new $k$ cluster $C$ such that, while minimizing clustering costs (focusing on $k$-centers), the number of reassigned points does not exceed a budget $b$ compared to a given historical cluster $H$.

**Strengths:**

This paper introduces the problem of label-consistent k-clustering, a novel clustering formulation that aims to optimize clustering cost while ensuring a consistency constraint, in the form of a maximum number of data point re-labelings, from a historical clustering. Meanwhile, it presents two constant-factor approximation algorithms for the k-center variant of the proposed problem.

**Weaknesses:**

1. The innovation is insufficient. The label consistency constraint of historical cluster H is highly similar to the existing "elastic clustering" model, which lacks fundamental innovation and is more like an incremental adjustment of the existing model for a specific scenario.

2. The core theory and approximation algorithm in the paper are only for the k-center objective function, but the k-median and k-mean are more practical in the evolutionary data scenarios mentioned by the authors. This limitation greatly weakens the general value of the model.

3. The asymptotic complexity of the polynomial-time algorithm includes an $O(n^2)$ term, which is too inefficient for large datasets (where n is very large) common in evolutionary data streams, making it impractical.

4. The OVERCOVER algorithm, with the strongest theoretical guarantee (close 2-approximation), is exponentially time-consuming ($O(2^k)$) on $k$. The authors also acknowledge that it is impractical for large $k$ values, making the optimal theoretical result impractical.

5. The construction of the historical clustering $H$ in the experimental setup is too artificial and fails to effectively simulate real evolutionary scenarios; furthermore, the baseline algorithm ignores the crucial budget constraint $b$ in the comparison.

**Questions:**

See the Weakness.

---

> ### Author Response · Authors · 2025-11-21
>
> We thank the reviewer for the thorough review and constructive comments.
>
> **Regarding W1:** We respectfully disagree with the reviewer’s assessment that the paper lacks innovation. Other reviewers explicitly highlighted the novelty and significance of our contributions, noting the new formulations, the new algorithms, and the theoretical insights we introduce, as well as the strength of our experimental evaluation.
>
> Regarding elastic clustering,  we are not aware of any prior formulation that is comparable to the one we propose. We are aware of work [1] related to fuzzy clustering, where points can fractionally belong to different clusters, and where ``elastic'' refers to the variable length of the encoding scheme [2]. Neither setting is close or related to our formulation. We would greatly appreciate if the reviewer could point to specific references so that we can address the concern more concretely in the revision.
>
> [1] https://pmc.ncbi.nlm.nih.gov/articles/PMC5730165/
>
> [2] https://pubmed.ncbi.nlm.nih.gov/31613788/
>
> **Regarding W2:** We believe that label-consistent $k$-center is an interesting problem in its own right and naturally captures tasks such as data summarization in evolving datasets. That said, we fully agree that label-consistent $k$-means and $k$-median would also be compelling and practically relevant extensions. We explicitly pose these as open problems in Section 6.
>
> **Regarding W3:** In practice, the distance aspect ratio $\Delta$ is often polynomially bounded in $n$, the number of points. In such settings, it is standard to speed up the guessing of $r^*$ by discretizing all distances into powers of $(1+\epsilon)$, for a small $\epsilon>0$. This reduces the number of candidate radii from $n^2$ to $O(\log n/\epsilon)$, at the cost of introducing only a multiplicative $(1+\epsilon)$ factor in the approximation. We will add this explanation in the revised version.
>
> **Regarding W4:**
> The algorithm OVERCOVER is not intended to be a practical method; rather, its purpose is illustrative. It highlights the tradeoff between approximation quality and runtime. Our primary motivation for including it is that it serves as a conceptual warm-up to our main algorithm, which is significantly more efficient while losing only an additional factor in the approximation (see also our response to W2 of reviewer hJxW).
>
> **Regarding W5:**  We describe how we handle evolutionary settings in lines 357--361, which follows the method described in the common answer on handling evolving point sets, where each timestep $X_t$ uses $C_{t-1}$ as historical clustering. This construction of historical clustering is natural and used in other approaches [3]. For simplicity, we constrained each timestep to have the same number of points, but point additions and deletions can be handled using the approach described in the common answer. We would appreciate clarification on why the reviewer believes our experimental setup fails to simulate real evolutionary scenarios, so that we can better address this issue.
>
>
> Furthermore, a key novelty of our algorithm is its ability to *take the budget parameter as part of the input* and to *provably respect the budget constraint on labels*. To the best of our knowledge, no prior algorithm enforces such a hard label budget, and the closest related notion is resilient clustering, which is why we chose it as a baseline. As our experiments indicate, the solution quality of the resilient clustering algorithm is significantly worse than ours—even though it does *not* enforce any hard constraint on the number of updates.
>
>
> [3] https://dl.acm.org/doi/abs/10.1145/1281192.1281212

---

### Author Response · Authors · 2025-11-21
**Common Answer**

We thank the reviewers for their constructive feedback. We begin by addressing the common questions, and then respond to the specific concerns raised in each review individually


**Handling evolving point sets.**
We elaborate on our writeup (lines 50--57) from the paper on how to handle evolving point sets.
Consider an evolving point set from $X_t$ to $X_{t+1}$, where we already have a clustering $C_t$ for $X_t$. Now, we construct an instance of our *label-consistent problem* by creating the historical clustering as follows.

For clarity, we first explain the construction when $X_{t+1}$ only contains new points (i.e., no points were deleted from $X_t$). In this case, we obtain the historical clustering $C\prime_t$ by retaining the assignments for points in $X_t$, and assigns each new point in $X_{t+1}$ to a closest center from $C_t$. Then, we seek for a label-consistent clustering $C_{t+1}$ of $C\prime_t$ that relabels at  most $b$ points in $X_{t+1}$.

Now, if $X_{t+1}$ deletes some points from $X_t$, we can simply remove them from $C_t$ to obtain the historical clustering $C\prime_t$, provided they were not centers in $C_t$. Otherwise, for each point $p$ deleted from $X_{t+1}$ that was a center in $C_t$, we select another point from $C_t$ (that is present in $X_{t+1}$) as a new center of the cluster. This ensures that all points in the cluster centered at $p$ in $C_t$ are still present together in a cluster with a new center. Note that if all the points of the cluster centered at $p$ in $C_t$ are deleted in $X_{t+1}$, then we can simply remove the whole cluster for $C\prime_t$. Thus, our model is perfectly applicable in this scenario as well, since all these points (whose center got deleted) still have a same (new) label, and
we want to compete with optimal solution that relabels at most $b$ points from $X_{t+1}$.
We will elaborate on handling both data additions and data deletions in the revised version.

**Comparison to dynamic setting.**
Although label-consistent clustering shares some similarities to problems in dynamic setting, we believe that our problem is conceptually much closer to consistent or resilient clustering. In these latter settings, a **hard constraint** limiting the number of updates, and algorithms are evaluated against solutions that also satisfy the hard constraint. In contrast, in the dynamic setting algorithms typically aim to **minimize the number of updates (recourse)** while still competing against **all (offline) solutions**, without imposing such a constraint. Furthermore, our problem is motivated by a regime where the number of points is extremely large compared to the number of time steps in which the input changes. This stands in contrast to typical dynamic settings, which often assume many updates and therefore require sublinear or near-sublinear running times. This difference justifies why algorithms for our problem may run in time exceeding the sublinear bounds standard in dynamic clustering.

We will upload the revised manuscript soon.

---

### Author Response · Authors · 2025-12-02

We again thank all reviewers for their constructive and valuable feedback. Based on the suggestions, we have substantially revised and improved our manuscript and uploaded a new version in which all additions and major changes are highlighted in blue.

Below we summarize the paper's merits as highlighted by the reviewers and the key changes made in this revision.

**Merits of the paper**

- Our problem setting is interesting and well-motivated (Reviewers 2CHi and KFw2)

- Our algorithms are theoretically sound and with convincing guarantees  (Reviewers hJwX and KFw2)

- Our empirical evaluation strengthens the technical analysis (Reviewers 2CHi, hJwX and KFw2)

- The paper is well-written (Reviewer hJwX)


**Key changes**

- Additional motivation of the necessity of a hard bound on the recourse (reviewer hJwX): We have added further motivation of our setting and its comparison with *dynamic* setting in the introduction and the related work, respectively.

- How does the method deal with point additions and deletions (reviewers 2CHi and KFw2): We explain this in the revised introduction.

- Method used in experiments does not have approximation guarantee (reviewer hJwX): This was a misunderstanding of the reviewer. We clarify this at the end of Section 5.1.

- Extended literature review (reviwer KFw2):  We added additional discussion on consistent (low-recourse) algorithms and dynamic clustering in Section 2.


- Clarification on the role of $b$ (reviewer 2CHi and KFw2): We have provided additional explanation on the role of $b$ in Section 3. Moreover, we explain how it is used in our algorithms in Section 4.3.

- How to guess the optimal radius and what is the computation cost for guessing (reviewer KFw2): We elaborate on this in Section 4.2.

We believe these revisions address all reviewer concerns while strengthening our paper substantially.

---

### Meta-Review · Area_Chair_B1G6 · 2025-12-28

**Summary:**

The reviews are overall negative. The concerns raised by the reviewers are mostly valid. The authors tried to address the concerns, but I think they are not quite to the point. I suggest to reject the paper.

**Reviewer Concerns:**

I identify the following as major concerns.

1. The evolving data setting is not discussed in a formal way, but is the main claim of the paper.
2. The reviewers find the modeling of the problem, especially the benchmark for the consistency, not natural.

For 1, I still consider this not resolved. While the authors tried to add a paragraph to explain the evolving data setting, I don't find how the theory/algorithms/analysis are changed accordingly.

For 2, the authors tried to emphasize their own setting, but I think what the reviewers mentioned also makes sense, and it is expected that the authors give some discussion on the other settings. This is not well addressed.

The other concerns seem to be minor and are adequately addressed by the authors.

**Reviewer Scores:**

Reviewer dpc9 may slightly increase the score, but probably still leaning to rejection. Other reviewers will not change the score.

---

### Decision · Program_Chairs · 2026-01-26

Reject